# BlurGuard: A Simple Approach for Robustifying Image Protection Against AI-Powered Editing

Jinsu Kim[1]* Yunhun Nam[1]* Minseon Kim[2] Sangpil Kim[1] Jongheon Jeong[1]

[1]Korea University [2]Microsoft Research Montréal

*{tonmmy222, yh0326}@korea.ac.kr

## Abstract

Recent advances in text-to-image models have increased the exposure of powerful image editing techniques as a tool, raising concerns about their potential for malicious use. An emerging line of research to address such threats focuses on implanting ("protective") adversarial noise into images before their public release, so future attempts to edit them using text-to-image models can be impeded. However, subsequent works have shown that these adversarial noises are often easily "reversed," *e.g.*, with techniques as simple as JPEG compression, casting doubt on the practicality of the approach. In this paper, we argue that adversarial noise for image protection should not only be *imperceptible*, as has been a primary focus of prior work, but also *irreversible*, *viz.*, it should be difficult to detect as noise provided that the original image is hidden. We propose a surprisingly simple method to enhance the robustness of image protection methods against noise reversal techniques. Specifically, it applies an adaptive per-region Gaussian blur on the noise to adjust the overall frequency spectrum. Through extensive experiments, we show that our method consistently improves the per-sample worst-case protection performance of existing methods against a wide range of reversal techniques on diverse image editing scenarios, while also reducing quality degradation due to noise in terms of perceptual metrics. Code is available at https://github.com/jsu-kim/BlurGuard.

## 1 Introduction

Generative AI has been revolutionizing computer vision with its remarkable capabilities in visual synthesis across complex domains, including images [4, 37, 67], 3D models [16, 52, 70], and videos [8, 69]. *Text-to-image models* [23, 67, 75], powered by large-scale diffusion-based generative models [33, 73, 85], are one of the representative examples of current generative AI, given the significant interest they have garnered within the research community. For example, these models have facilitated diverse application research, *e.g.*, personalizing the models into specific artistic styles [26, 74], inpainting or editing an image based on textual prompts [55, 58], and even following detailed instructions [7], to name a few. Publicly available text-to-image models, such as Stable Diffusion (SD) [23, 67], have further expanded the accessibility of these techniques to broader audiences.

Despite advancements, the broader accessibility of text-to-image models has also raised serious concerns about their potential for misuse. Specifically, natural-looking manipulation of an image can produce harmful or deceptive content, resulting in fake news, violations of copyright, publicity or even portrait rights. For instance, current models can easily replicate an artist's style without permission or infringe on portrait rights by manipulating images of any victims. Such malicious editing is particularly concerning as the quality of image generation approaches a level that is nearly indistinguishable from reality; this "realism" of generative AI increases the risks of misinformation,

---

*Equal contribution.

39th Conference on Neural Information Processing Systems (NeurIPS 2025).

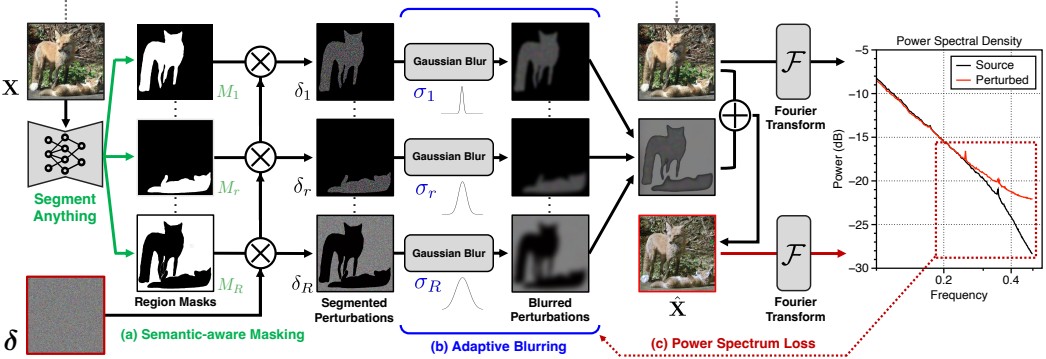

Figure 1: Overview of *BlurGuard*, a novel framework for constructing robust image protection. We **(a)** leverage Segment Anything [40] to obtain semantic-aware region-wise perturbations, **(b)** apply different levels of Gaussian blur, and **(c)** update blur-intensity parameters to minimize the power-spectrum gap before and after adding the perturbation.

harassment, intellectual property theft, and privacy violations, as manipulated images can spread false narratives, damage reputations, or exploit individuals without consent.

To mitigate these risks, recent efforts [77, 80, 90, 101] have focused on developing technical strategies to raise the cost of malicious image editing, which has been significantly lowered by generative AI. One of the prominent ideas in this direction is to apply *adversarial noise* [29, 87] to images before they are exposed to potential malicious uses [48, 49, 77, 96], ensuring that the noise is (a) imperceptible enough to preserve the original quality of the given image, and (b) capable of completely disrupting the behavior of generative AI. For example, Salman et al. [77] have shown that such noise can effectively prevent an image from being edited using off-the-shelf text-to-image models, such as Stable Diffusion.

However, later studies [34, 102] have raised concerns about the reliability of adversarial attack-based protection methods for practical use. Specifically, they found that all the existing methods are vulnerable to simple noise removal techniques, such as JPEG compression [91] and diffusion-based noise purification [63, 66], effectively neutralizing the protections and allowing attackers to bypass them. In other words, adversarial image protection is often easily "reversible." These observations suggest that *simply finding imperceptible noise is not sufficient* for withstanding diverse noise removal techniques. This is intuitively reasonable, as imperceptibility is defined by human perception, whereas existing noise removal methods are not necessarily designed to align with humans.

**Contributions**   In this paper, we move away from the previous focus on imperceptibility as the primary criterion of crafting adversarial noise. Instead, we argue that adversarial noise should also minimally affect the original distribution of images across different representations, ensuring that it remains undetectable by detectors based on unforeseen image representations. We start by observing that adversarial attacks optimized for imperceptibility, *e.g.*, by constraining the noise within a small $\ell_\infty$-ball, often produce noise that becomes easily detectable when viewed in the frequency spectrum. We develop this observation and show that existing adversarial attack methods for image protection commonly suffer from this issue, explaining why they can be easily bypassed.

To further support our claim, we propose a surprisingly simple method that improves the robustness of adversarial attacks against noise removal by minimizing their detectability in the frequency domain. Specifically, we introduce (a) a learnable Gaussian blurring layer to effectively limit the frequency bands of adversarial noise for each semantic region of a given image, and (b) a novel objective to optimize the blurring parameters based on the power spectrum of the perturbed image (see Figure 1). In this way, the adversarial noise can automatically adjust its frequency bands to maintain high naturalness to withstand purification techniques.

We perform an extensive evaluation of our method in comparison to existing adversarial attack based image protection methods [48, 49, 77, 96], under a challenging scenario where a malicious user selects the best-case performance after diverse state-of-the-art noise removal techniques [34, 63, 102]. As an attempt to standardize the evaluation, we also construct a new curated dataset for assessing the performance of image protection techniques against malicious editing, addressing the current lack of

a public benchmark. Our results show that the proposed method, despite its simplicity, significantly improves the per-sample worst-case protection performance on the benchmark, while effectively controlling its imperceptibility as well. For example, our method could maintain the effectiveness in FID up to $93\%$ after all purification tested, while the previous best does only $48\%$.

## 2   Related Work

**Protecting images against generative AI**   The concept of embedding adversarial noise to protect images from unauthorized AI systems has been introduced in various contexts [17, 77, 79, 80, 90]. For example, Shan et al. [79] considered scenarios in which a "tracker" trains a facial recognition model using online photos of specific users, and proposed leveraging adversarial noise to disrupt model training and protect user privacy. Salman et al. [77] considered similar threats in the context of text-to-image model based image editing pipelines; Shan et al. [80] and Van Le et al. [90] have focused on a specific setup of *style mimicry*, where a malicious user further fine-tunes a text-to-image model to extract artistic styles in images; Choi et al. [18] have focused on *inpainting*, which modifies specific regions of human images. The robustness of image protection has only recently been explored [15, 92], but existing methods remain limited to handling a single type of editing task (*e.g.*, inpainting) and are tailored to naïve purification techniques such as JPEG. In this paper, we propose a simple and versatile method that generalizes across a broader range of image editing tasks, with a focus on the more challenging scenario of worst-case purification techniques.

**Frequency-based adversarial examples**   Some early works constrained their adversarial perturbations within low-frequency bands for several reasons, including reducing computational cost [31], enhancing robustness [81], and improving imperceptibility [2, 13, 56, 59, 95]. In the context of generative diffusion models, Ahn et al. [1] have recently leveraged the frequency domain to enhance the imperceptibility of adversarial perturbation but they do not focus on the robustness of protection. In image watermarking, Liu et al. [51] have considered influence functions with frequency constraint to generate adversarial watermarks against personalized diffusion models, showing robustness against simple compression techniques such as JPEG and WebP. Some works [54, 35] considered low-frequency watermarks so that they remain decodable after editing. Beyond these works, we propose a simpler yet versatile frequency-based approach for robust image protection through a novel power spectrum regularization, targeting wider ranges of tasks and purification methods.

We provide more discussion of related work in Appendix A.1.

## 3   Preliminaries

**Text-to-image diffusion models**   *Diffusion models* [22, 33, 84] learn a data distribution $p_{\text{data}}(\mathbf{x})$ through denoising interpolants between $p_{\text{data}}(\mathbf{x})$ and Gaussian noise. *Text-to-image diffusion models* [4, 73, 75] incorporate texts as external variables for conditional generation. From an architectural perspective, existing large-scale diffusion models typically employ one of two designs to handle high-resolution inputs: (a) *latent diffusion models* (LDMs) [73], which first map $\mathbf{x}$ into a latent space of lower-resolution via an encoder $\mathbf{z} := \mathcal{E}(\mathbf{x})$, and (b) *cascaded diffusion models* [76], which first train a low-resolution pixel diffusion model, followed by progressive super-resolution modules.

**Adversarial image protection**   In response to the emerging threats of malicious image editing enabled by generative AI, recent efforts [48, 49, 77, 96] have utilized adversarial examples [5, 12, 87] as a protection scheme of images against public generative models, *e.g.*, text-to-image diffusion models. In particular, PhotoGuard [77] proposes two different ways of disrupting the inner working of LDMs (*e.g.*, Stable Diffusion) through adversarial noise: either (a) by corrupting the encoder model within LDM ("*encoder attack*"), or (b) by making its denoising process more difficult ("*diffusion attack*"). For example, the encoder attack considers the following optimization given $\mathbf{x}$:

$$\hat{\mathbf{x}}_{\text{adv}} = \underset{d(\hat{\mathbf{x}}, \mathbf{x}) \leq \epsilon}{\arg\min}\, L_{\text{enc}}(\hat{\mathbf{x}}), \quad \text{where} \quad L_{\text{enc}} := ||\mathcal{E}(\hat{\mathbf{x}}) - \mathbf{z}_{\text{target}}||_2^2, \qquad (1)$$

where $d(\cdot, \cdot)$ is a distance metric, and $\mathbf{z}_{\text{target}}$ is a certain target latent representation, *e.g.*, that of a gray image. In practice, such an optimization is performed via *projected gradient descent* (PGD) [57], using a distance metric that allows easy projection while maintaining imperceptibility, such as $\ell_\infty$ distance $d(\hat{\mathbf{x}}, \mathbf{x}) := ||\hat{\mathbf{x}} - \mathbf{x}||_\infty$:

$$\hat{\mathbf{x}} \leftarrow \text{Proj}_{||\hat{\mathbf{x}} - \mathbf{x}||_\infty \leq \epsilon} \left( \hat{\mathbf{x}} - \gamma \cdot \text{sign}(\nabla_{\hat{\mathbf{x}}} L_{\text{enc}}(\hat{\mathbf{x}})) \right), \qquad (2)$$

where $\gamma$ is the step-size for each update.

**Adversarial purification**  The idea of neutralizing adversarial noise through *adversarial purification* methods has been repeatedly explored over time; examples include JPEG compression [30], GAN-based projection [78], and diffusion model-based denoising [63], among others. Yet, later works [3, 45] have shown that these approaches are often insufficient as a thorough defense mechanism. Given the "flipped" attack-defense relationship introduced by the new setup of adversarial protection, however, these purification methods have now become strong "attack" mechanisms [34, 102]; here, the key difference is that adversarial noise (intended for protection) no longer has the opportunity to adapt to future purification methods, even if there are ways to bypass individual purification schemes. Ultimately, we question: **"Can a single adversarial noise be resilient to *all* possible purification methods, offering *irreversible* protection against evolving noise reversal techniques?"**

## 4   Method

In approaching the above question, we argue that irreversible adversarial noise should have only minimal impact on the *data likelihood* of the original image, so that no specific purification method can project the noise back onto the given manifold defined by $p_{\text{data}}(\mathbf{x})$. We motivate this intuition by showing that existing adversarial protections, even when imperceptible in the pixel space, are often easily detectable when analyzed in their frequency bandwidth, and consequently by purification methods (Section 4.1). Based on this observation, we propose BlurGuard, a simple method to enhance robustness of adversarial protection by adaptively restricting the frequency bands of perturbation to align more closely with the natural frequency characteristics of the data, making the noise less distinguishable from the underlying distribution $p_{\text{data}}(\mathbf{x})$ (Section 4.2 and 4.3).

**Threat model**  Consider a private image $\mathbf{x}$ sampled from $p_{\text{data}}(\mathbf{x})$ (*i.e.*, $\mathbf{x}$ initially has a high data likelihood), which is accessible only to a *protector* (or "defender"). The goal of the protector is to find an adversarial example $\hat{\mathbf{x}}$ near $\mathbf{x}$ such that unauthorized editing of $\hat{\mathbf{x}}$ using a given text-to-image diffusion model $p_\theta(\mathbf{x}, c)$ becomes difficult. In order to control the imperceptibility of $\hat{\mathbf{x}}$, we assume that $\hat{\mathbf{x}}$ is bounded within a certain ball $B_\epsilon(\mathbf{x})$ around $\mathbf{x}$, where its definition depends on the chosen distance metric $d(\cdot, \cdot)$:

$$B_\epsilon(\mathbf{x}) := \{\mathbf{x}' : d(\mathbf{x}', \mathbf{x}) \leq \epsilon\}. \tag{3}$$

There can be multiple ways for $\hat{\mathbf{x}}$ to disrupt the behavior of $p_\theta(\mathbf{x}, c)$, *e.g.*, by corrupting either the encoder or denoiser within LDMs, as done in PhotoGuard. We denote the protector's adversarial objective as $L_{\text{adv}}(\hat{\mathbf{x}}; p_\theta)$.

From the perspective of an *unauthorized editor* (or "attacker"), on the other hand, the goal is now to neutralize a given protected image $\hat{\mathbf{x}}$ by purifying the added adversarial noise, *i.e.*, to recover $\mathbf{x}$ from $\hat{\mathbf{x}}$. Given the common information that $\mathbf{x}$ has a high likelihood with respect to $p_{\text{data}}$, existing purification methods essentially leverage specific prior knowledge about $p_{\text{data}}$ and aim to maximize $p_{\text{data}}(\hat{\mathbf{x}})$. For example, JPEG compression [91] is designed to remove less-frequent signals in natural scenes. In principle, the attacker has no restriction on the choice of purification methods to apply, as long as $\mathbf{x}$ remains hidden. This means that, an "oracle" attacker, equipped with a close approximation of $p_{\text{data}}$, could even directly maximize $p_{\text{data}}(\hat{\mathbf{x}})$.

**Need for "naturalistic" protection**  Ultimately, incorporating the attacker's capabilities, we consider the following minimax objective to find adversarial protection:

$$\hat{\mathbf{x}}_{\text{adv}} := \underset{\hat{\mathbf{x}} \in B_\epsilon(\mathbf{x})}{\arg\min} \; L_{\text{adv}}(\text{purify}(\hat{\mathbf{x}}); p_\theta), \text{ where } \text{purify}(\hat{\mathbf{x}}) := \underset{\mathbf{y} \in B_\epsilon(\hat{\mathbf{x}})}{\arg\max} \; \log p_{\text{data}}(\mathbf{y}, c). \tag{4}$$

The objective (4) suggests that the optimal protection $\hat{\mathbf{x}}_{\text{adv}}$ should lie among the points in $B_\epsilon(\mathbf{x})$ where the data likelihood is at least $p_{\text{data}}(\mathbf{x})$; otherwise, $\text{purify}(\hat{\mathbf{x}})$ could identify $\mathbf{x}$ as a feasible point in the inner optimization, thus succeeding in reversing the protection.

### 4.1   A Frequency View of Adversarial Image Protection

We next question whether the existing methods for adversarial protection [49, 48, 77, 96] are capable of generating sufficiently natural perturbation, *i.e.*, ones that can effectively solve (4). In general, the

naturalness of adversarial examples has been controlled by constraining the noise to meet a certain *imperceptibility* criterion, *e.g.*, being within a small $\ell_p$-ball; current adversarial protection techniques also follow this approach.

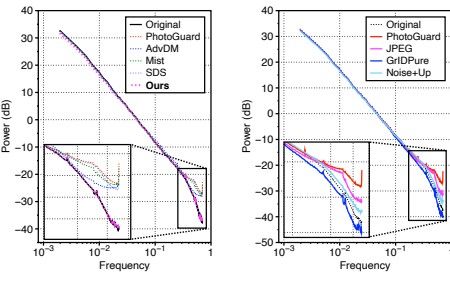

(a) Attacked images     (b) After purification

Figure 2: Comparisons of RAPSD on ImageNet-Edit (Appendix C.3) with SD-v1.4.

However, pursuing imperceptibility does not necessarily ensure naturalness; rather, imperceptibility is a specific instance of naturalness, when perception is limited to human vision. Given that diffusion models reconstruct images by progressively moving from low to high frequencies [72], it is natural to expect that the frequency-spectrum naturalness of an image also influences its data likelihood $p_{\text{data}}$. In Figure 2, we indeed observe that an imperceptible, $\ell_\infty$-constrained image protection from various existing methods often produces a more significant deviation when viewed in its *radially-averaged power spectral density* (RAPSD), making it difficult to consider as a natural scene; *e.g.*, one that follows the $1/f^2$-law [9].

## 4.2 BlurGuard: Robust Adversarial Protection via Adaptive Gaussian Blurring

Motivated by Section 4.1, we develop BlurGuard, a simple plug-and-play approach to improve the robustness of image protection methods by regularizing the power spectrum of adversarial noise. Overall, we generally observe the bias of existing image protection techniques towards high-frequency bands and aim to close this spectrum gap (from natural images) by applying "spatially-adaptive" Gaussian blur to the noise. In this way, BlurGuard can selectively allocate high-frequency noise to regions that have less impact on the overall spectrum (*e.g.*, as shown in Figure 2a), thus maintaining strong protection while appearing more natural.

**Learnable Gaussian blurring** To effectively control the frequency spectrum of image protection, we apply a Gaussian blur to adversarial noise with a learnable blur intensity $\sigma > 0$. That is, given an input $\mathbf{x}$, we start by parameterizing its protection $\hat{\mathbf{x}}$ as follows:

$$\hat{\mathbf{x}} := \mathbf{x} + \mathcal{G}(\boldsymbol{\delta}; \sigma), \tag{5}$$

where $\mathcal{G}$ represents the Gaussian blur operation, and $\boldsymbol{\delta}$ is a learnable parameter. For image-like inputs, Gaussian blur can be implemented as a depthwise 2D convolution,[2] where the kernel matrix $H_{\sigma,k} \in \mathbb{R}^{(2k+1)\times(2k+1)}$ is defined by:

$$H_{\sigma,k}(u,v) := G(u)\,G(v) \ \text{ for } u,v \in \{-k,\ldots,k\}, \text{where} \ \ G(z) := \frac{1}{\sigma\sqrt{2\pi}}\exp\left(-\frac{z^2}{2\sigma^2}\right). \tag{6}$$

The convolution in (6) effectively serves as a low-pass filter; it is equivalent to modulating the Fourier-transformed $\mathbf{x}$ by a Gaussian function, given that the Fourier transform of Gaussian kernel is again Gaussian. Here, the blur intensity $\sigma$ determines the cut-off frequency. We initialize $\omega := \log \sigma$ as learnable parameters instead of directly optimizing $\sigma$, primarily to avoid numerical instability.

**Power spectrum regularization** Given an image $\mathbf{x}$ and its protection $\hat{\mathbf{x}}$, we aim to optimize the blur intensity $\sigma$ of Gaussian blur to minimize their discrepancy in terms of frequency spectrum. To this end, we apply Fast Fourier Transform (FFT) to both $\mathbf{x}$ and $\hat{\mathbf{x}}$, and compute RAPSD of them. Specifically, for a 2D Fourier frequency matrix $\mathbf{f} \in \mathbb{C}^{hw}$, we first partition the coordinates of $\mathbf{f}$ into $B$ bands with respect to the radius from the center; yielding $B$ sets of coordinates $\{\mathcal{B}_b\}_{b=1}^{B}$. Then, RAPSD is defined by a $B$-dimensional vector, where the $b$-th entry is the squared magnitude of the FFT coefficients in $\mathcal{B}_b$:

$$\text{RAPSD}_b(\mathbf{f}) := \frac{1}{|\mathcal{B}_b|} \sum_{(u,v) \in \mathcal{B}_b} \left|\mathbf{f}(u,v)\right|^2. \tag{7}$$

---

[2]With this implementation, one can optimize $\sigma$ using conventional automatic differentiation libraries (*e.g.*, PyTorch), whereas standard implementations often do not support this.

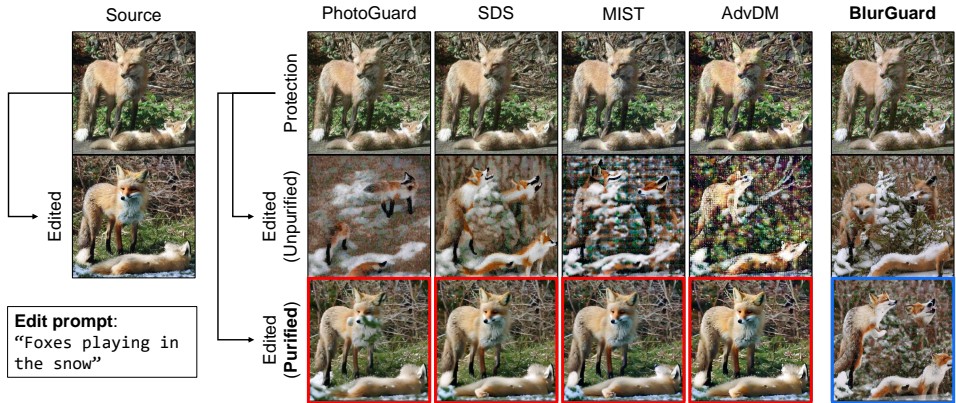

Figure 3: Qualitative comparison of image-to-image generation results ($\epsilon = \frac{16}{255}$). Before purification, all protection methods effectively safeguard the images. However, in the worst-case scenario, the baselines often fail to maintain their protection, producing outputs nearly identical to the original edit. In contrast, BlurGuard continues to generate unrealistic images that do not resemble either the source image or the generation from it. More qualitative results can be found in Figure 12 in Appendix.

Once the frequency matrices $\mathbf{f}$ and $\hat{\mathbf{f}}$ are obtained from $\mathbf{x}$ and $\hat{\mathbf{x}}$, respectively, we define a new *power spectrum regularization* as follows:[3]

$$L_{\text{freq}}(\hat{\mathbf{x}}, \mathbf{x}) := \left\| \log \frac{\text{RAPSD}(\hat{\mathbf{f}})}{\text{RAPSD}(\mathbf{f})} \right\|_\infty \tag{8}$$

where $\|z\|_\infty := \max_i |z_i|$ is the $\ell_\infty$ norm. By minimizing (8), one can ensure that the spectral gaps between $\mathbf{x}$ and $\hat{\mathbf{x}}$ are within a certain margin for all frequencies. BlurGuard uses this regularization primarily to adjust the optimal blur intensity parameter $\boldsymbol{\sigma}$.

**Per-region adaptation** Considering that an image is composed of multiple regions with diverse patterns, it is reasonable to divide the image into several semantic regions and apply different blur intensities; so that the overall frequency characteristics of the perturbed image remain better preserved. To achieve this, we leverage Segment Anything Model (SAM) [40], one of recent image segmentation foundation models, to obtain $R$ binary segmentation masks $\{M_r\}_{r=1}^R$ out of a given image $\mathbf{x}$. Next, we introduce per-region blur intensity parameters $\{\sigma_r\}_{r=1}^R$, and they are jointly optimized through the following aggregation combined with $\{M_r\}_{r=1}^R$ ($\odot$ is the Hadamard product):

$$\hat{\mathbf{x}} = \mathbf{x} + \sum_{r=1}^R M_r \odot \mathcal{G}(\boldsymbol{\delta}; \sigma_r). \tag{9}$$

### 4.3 Overall Objective

Given an image $\mathbf{x}$, BlurGuard jointly optimizes the perturbation parameters $\boldsymbol{\delta}$ and blur intensities $\boldsymbol{\sigma} := \{\sigma_r\}_{r=1}^R$, and returns the resulting protection $\hat{\mathbf{x}}$ defined as (9). To optimize $\boldsymbol{\delta}$, we combine our proposed power spectrum regularization $L_{\text{freq}}$ (8) with an existing adversarial protection objective, say $L_{\text{adv}}$, given a certain adversarial budget; here, we consider an $\epsilon$-ball in $\ell_\infty$ distance, following standard practice:

$$\boldsymbol{\delta} = \underset{\|\boldsymbol{\delta}\|_\infty \leq \epsilon}{\arg\min} \Big( L_{\text{adv}}(\hat{\mathbf{x}}, \mathbf{x}) + \lambda \cdot L_{\text{freq}}(\hat{\mathbf{x}}, \mathbf{x}) \Big), \tag{10}$$

where $\lambda > 0$ is a hyperparameter.[4] We use iterative optimization (*viz.*, PGD) for solving (10). A slight modification to (2) is that we do not apply $\text{sign}(\cdot)$ for projecting gradients; instead, we simply use $\ell_2$-normalization. This is to reduce gradient noise that can arise from sign-based projection, given that the blurring operation already alters the original gradient direction.

---

[3]We average over the channels before applying FFT on $\mathbf{x}$ and $\hat{\mathbf{x}}$, ensuring that their frequency matrices remain two-dimensional.

[4]We fix $\lambda = 10$ throughout all the experiments in this paper.

In terms of $\sigma$, on the other hand, we observe that optimizing $\sigma$ for $L_{\text{adv}}$ can be in conflict with $L_{\text{freq}}$, causing it to converge to a looser frequency band and thereby affecting robustness. As a simple remedy, we first optimize $\sigma$ with respect to the power spectrum objective only, *i.e.*, for minimizing $L_{\text{freq}}(\hat{x}, x)$, and freeze the learned $\sigma$ throughout the optimization of $\delta$ as in (10). The detailed procedure is provided in Algorithm 1 in the Appendix.

**Adversarial objective**    In principle, our proposed adaptive blurring scheme is versatile and compatible with any form of adversarial objective $L_{\text{adv}}$ that aims to protect an image $x$ from being edited. In our experiments, we adopt the encoder loss $L_{\text{enc}}$ of PhotoGuard (1), primarily for simplicity. Here, we take $z_{\text{target}} = 0$, also following Salman et al. [77].

## 5    Experiments

To verify the effectiveness of BlurGuard, we perform extensive evaluation covering wide AI-powered editing scenarios, including image-to-image generation [73], inpainting-based editing [55], instruction-based editing [7], textual inversion [26], and DreamBooth [74].[5] We compare various protection methods applicable to editing tasks, considering a fixed $\ell_\infty$-ball of $\epsilon = \frac{16}{255}$ as the adversarial budget following the standard. Our evaluation covers both (a) *naturalness*: how imperceptible the protection is, and (b) *effectiveness*: how much a protected image can deviate from the original image when both are edited. The detailed experimental setups, *e.g.*, datasets, baselines, evaluation metrics, hyperparameters, *etc.*, can be found in Appendix C.

**Purification methods**    To assess the robustness of each protection method, we consider diverse noise purification techniques, including: **(a)** *JPEG compression*, **(b)** JPEG followed by image upscaling [62], and **(c)** *Noisy upscaling* [34], which combines Gaussian noise and upscaling for a more aggressive purification. More advanced diffusion-based purification methods are also considered; including **(d)** Impress [10], **(e)** DiffPure [63], **(f)** GrIDPure [102], and **(g)** PDM-Pure [94]. Details on purification methods can be found in Appendix C.5.

**Worst-case effectiveness**    Unless otherwise noted, we mainly focus on *per-sample worst-case effectiveness* of image protection methods, which compute the "lowest" performance scores after applying all the purification techniques available (*i.e.*, from **(a)** to **(g)**) individually for each sample. This comparison assumes an "oracle" attacker equipped with many noise purification techniques, which is essentially closer to more realistic threats.

### 5.1    Results

**Image-to-image generation**    We test BlurGuard for image-to-image editing based on Stable Diffusion (SD) v1.4 model [73], following previous setups.[6] To effectively evaluate image protection performance across diverse cases, we have curated a new, custom subset of 80 image samples taken from ImageNet [21], coined *ImageNet-Edit*; more details can be found in Appendix C.3.

Table 1 summarizes the results. Overall, we observe that BlurGuard consistently surpasses the baselines tested across all metrics, both in naturalness (up to 30% reduction in LPIPS) and in worst-case robustness (up to 9.8% increase in FID). This observation is further supported qualitatively in Figure 11a. We also remark the high variance of BlurGuard in PSNR naturalness; this is due to that BlurGuard assigns an adaptive amount of perturbation per-image to achieve high naturalness, which is an expected behavior.

In Table 3, we compare the ratio of effectiveness in FID before and after purification. Notably, BlurGuard retains 92.9% of its original protective efficacy even after purification; far surpassing all other methods, which fall between 38.5% and 48.4%. This confirms the superior robustness of BlurGuard to existing purification approaches.

**Inpainting**    Inpainting-based editing incorporates a binary mask information as well as a textual prompt, which can be useful for modifying specific regions (*e.g.*, the surrounding environment or

---

[5]We report the textual inversion and DreamBooth experiments in Appendix D.1.

[6]We also provide results with SD-v2.1, SD-v3.5, and FLUX.1-dev models in Appendix D.3.

| IN-Edit ($\epsilon = \frac{16}{255}$) | Naturalness | | | Worst-case Effectiveness | | | | |
|---|---|---|---|---|---|---|---|---|
| Method | LPIPS ↓ | SSIM ↑ | PSNR ↑ | FID ↑ | LPIPS ↑ | SSIM ↓ | PSNR ↓ | IA ↓ |
| PhotoGuard [77] | 0.34 ± 0.11 | 0.70 ± 0.11 | 28.2 ± 0.32 | 92.21 | 0.27 ± 0.07 | 0.73 ± 0.10 | 29.9 ± 1.07 | 0.94 ± 0.04 |
| AdvDM [49] | 0.36 ± 0.12 | 0.74 ± 0.09 | 28.9 ± 0.28 | 98.27 | 0.30 ± 0.08 | 0.73 ± 0.09 | 30.2 ± 1.17 | 0.93 ± 0.04 |
| Mist [48] | 0.35 ± 0.12 | 0.72 ± 0.10 | 28.6 ± 0.25 | 90.96 | 0.30 ± 0.07 | 0.71 ± 0.09 | 29.7 ± 0.86 | 0.93 ± 0.04 |
| SDS [96] | 0.30 ± 0.09 | 0.74 ± 0.09 | 29.3 ± 0.45 | 91.48 | 0.26 ± 0.07 | 0.73 ± 0.09 | 30.3 ± 1.23 | 0.94 ± 0.04 |
| **BlurGuard (Ours)** | **0.21 ± 0.10** | **0.93 ± 0.09** | **31.1 ± 2.29** | **107.88** | **0.32 ± 0.09** | **0.70 ± 0.09** | **28.8 ± 0.42** | **0.92 ± 0.05** |

(a) Image-to-image generation

| Helen ($\epsilon = \frac{16}{255}$) | Naturalness | | | Worst-case Effectiveness | | | | |
|---|---|---|---|---|---|---|---|---|
| Method | LPIPS ↓ | SSIM ↑ | PSNR ↑ | FID ↑ | LPIPS ↑ | SSIM ↓ | PSNR ↓ | IA ↓ |
| PhotoGuard [77] | 0.10 ± 0.05 | 0.91 ± 0.04 | 35.4 ± 1.71 | 131.93 | 0.34 ± 0.08 | 0.69 ± 0.08 | 29.5 ± 0.50 | 0.93 ± 0.04 |
| AdvDM [49] | **0.06 ± 0.04** | **0.97 ± 0.02** | **40.4 ± 2.29** | 105.26 | 0.25 ± 0.07 | 0.76 ± 0.08 | 30.2 ± 0.70 | 0.95 ± 0.03 |
| Mist [48] | 0.07 ± 0.04 | 0.95 ± 0.02 | 37.3 ± 1.62 | 109.37 | 0.27 ± 0.07 | 0.74 ± 0.07 | 29.9 ± 0.52 | 0.95 ± 0.03 |
| SDS [96] | 0.16 ± 0.07 | 0.92 ± 0.04 | 35.4 ± 1.75 | 133.78 | 0.34 ± 0.08 | 0.69 ± 0.08 | 29.6 ± 0.60 | 0.92 ± 0.03 |
| DiffusionGuard [18] | 0.09 ± 0.05 | 0.94 ± 0.03 | 36.6 ± 1.64 | 123.32 | 0.32 ± 0.08 | 0.71 ± 0.08 | 29.7 ± 0.54 | 0.93 ± 0.04 |
| **BlurGuard (Ours)** | 0.11 ± 0.05 | **0.97 ± 0.02** | 36.5 ± 2.66 | **162.92** | **0.45 ± 0.09** | **0.64 ± 0.09** | **29.0 ± 0.45** | **0.86 ± 0.05** |

(b) Inpainting

| MagicBrush ($\epsilon = \frac{16}{255}$) | Naturalness | | | Worst-case Effectiveness | | | | |
|---|---|---|---|---|---|---|---|---|
| Method | LPIPS ↓ | SSIM ↑ | PSNR ↑ | FID ↑ | LPIPS ↑ | SSIM ↓ | PSNR ↓ | IA ↓ |
| PhotoGuard [77] | **0.19 ± 0.10** | 0.74 ± 0.09 | **31.1 ± 0.22** | 96.23 | 0.21 ± 0.09 | 0.73 ± 0.11 | 30.7 ± 1.50 | 0.96 ± 0.03 |
| AdvDM [49] | 0.27 ± 0.08 | 0.76 ± 0.07 | 28.9 ± 0.24 | 123.39 | 0.30 ± 0.09 | 0.67 ± 0.13 | 30.2 ± 1.25 | 0.93 ± 0.04 |
| Mist [48] | 0.26 ± 0.09 | 0.74 ± 0.08 | 28.6 ± 0.17 | 123.54 | 0.30 ± 0.09 | 0.66 ± 0.12 | 29.8 ± 0.94 | 0.93 ± 0.04 |
| SDS [96] | 0.23 ± 0.07 | 0.74 ± 0.07 | 29.1 ± 0.37 | 115.23 | 0.27 ± 0.09 | 0.68 ± 0.13 | 30.3 ± 1.38 | 0.94 ± 0.04 |
| EditShield [15] | 0.36 ± 0.13 | 0.75 ± 0.09 | 31.0 ± 0.23 | 120.13 | 0.33 ± 0.09 | 0.65 ± 0.13 | 29.9 ± 1.26 | 0.93 ± 0.04 |
| **BlurGuard (Ours)** | 0.20 ± 0.08 | **0.89 ± 0.07** | 30.1 ± 1.76 | **138.22** | **0.36 ± 0.10** | **0.64 ± 0.12** | **28.8 ± 0.55** | **0.90 ± 0.06** |

(c) Instruction-based editing

Table 1: Comparisons of image protection within $\ell_\infty$-balls of $\epsilon = \frac{16}{255}$. *Naturalness* indicates how much a protected image deviates from the original. *Worst-case effectiveness* is measured after applying all purification methods to the protected image. The best is highlighted in **bold**, while the second-best in underlined.

| IGBench ($\varepsilon = \frac{16}{255}$) | Naturalness | | Worst-case Effect. (Seen) | | Worst-case Effect. (Unseen) | |
|---|---|---|---|---|---|---|
| Method | LPIPS ↓ | SSIM ↑ | FID ↑ | PSNR ↓ | FID ↑ | PSNR ↓ |
| PhotoGuard [77] | 0.03 ± 0.02 | 0.96 ± 0.02 | 103.83 | 37.2 ± 14.2 | 67.05 | 35.5 ± 8.2 |
| AdvDM [49] | 0.03 ± 0.02 | 0.97 ± 0.02 | 125.66 | 35.3 ± 10.3 | 71.06 | 35.4 ± 7.0 |
| Mist [48] | **0.02 ± 0.01** | 0.98 ± 0.01 | 112.23 | 37.1 ± 14.2 | 58.48 | 36.4 ± 9.8 |
| SDS [96] | 0.05 ± 0.03 | 0.96 ± 0.02 | 113.30 | 37.1 ± 14.2 | 83.91 | 35.6 ± 7.2 |
| DiffusionGuard [18] | 0.03 ± 0.02 | 0.97 ± 0.02 | 134.25 | 37.1 ± 14.2 | 61.32 | 35.9 ± 9.9 |
| **BlurGuard (Ours)** | 0.03 ± 0.02 | **0.99 ± 0.01** | **140.82** | **34.9 ± 17.0** | **90.32** | **35.3 ± 8.9** |

Table 2: Comparison of image protection for inpainting under mask variability. We test our method on InpaintGuardBench (IGBench) [18] that provides unseen shapes of mask not exposed during each protection phase. The best is highlighted in **bold**, while the second-best in underlined.

objects) while preserving others (*e.g.*, a person's face) Consequently, image protections should be confined to the facial region rather than applied to the entire image, and we follow this setup. For evaluation, we use the Helen dataset [44], following Cao et al. [10].

As reported in Table 1, BlurGuard again achieves the strongest worst-case effectiveness, while maintaining high naturalness; for example, under the same $\epsilon = \frac{16}{255}$ budget, BlurGuard finds perturbations that achieve 0.45 LPIPS on average in terms of worst-case effectiveness, which is 32.3% higher than the second-best baseline. Qualitative results can be found in Appendix E.

To further evaluate the effectiveness of BlurGuard in realistic inpainting scenarios with diverse editing masks, we assess its robustness under mask variability using the InpaintGuardBench [18]. Each image is tested with six masks, including both "seen" (used for protection) and "unseen" (novel) ones.

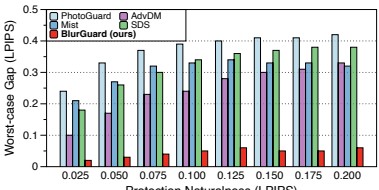

Figure 4: Comparison of protection naturalness *vs.* worst-case effectiveness (both in LPIPS) across different protection methods.

| ($\epsilon = \frac{16}{255}$) | **Purified?** | | **Robustness** |
|---|---|---|---|
| **Method** | ✗ (A) | ✓ (B) | (B/A; %) |
| PhotoGuard | 202.0 | 92.2 | 45.6 |
| AdvDM | **255.6** | 98.3 | 38.5 |
| Mist | 233.7 | 91.0 | 38.9 |
| SDS | 189.0 | 91.5 | 48.4 |
| **BlurGuard** | 116.2 | **107.9** | **92.9** |

Table 3: Comparison of protection effectiveness in FID before and after purification on ImageNet-Edit. Best result is marked in **bold**.

| Method | FID ↑ | LPIPS ↑ | SSIM ↓ |
|---|---|---|---|
| PhotoGuard | 134.0 | 0.36 | 0.66 |
| AdvDM | 150.8 | 0.40 | 0.69 |
| Mist | 147.2 | 0.40 | 0.68 |
| SDS | 142.6 | 0.36 | 0.68 |
| **BlurGuard** | **153.1** | **0.41** | **0.62** |

Table 4: Results of black-box image-to-image on ImageNet-Edit (SD-v1.4 → SD-v2.1). We use $\epsilon = \frac{16}{255}$ in $\ell_\infty$. Best result is marked in **bold**.

| IN-Edit ($\varepsilon = \frac{16}{255}$) | **Naturalness** | | | **Worst-case Effectiveness** | | | | |
|---|---|---|---|---|---|---|---|---|
| Method | LPIPS ↓ | SSIM ↑ | PSNR ↑ | FID ↑ | LPIPS ↑ | SSIM ↓ | PSNR ↓ | IA ↓ |
| AdvDM [49] | 0.36 ± 0.12 | 0.74 ± 0.09 | 28.9 ± 0.28 | 98.27 | 0.30 ± 0.08 | 0.73 ± 0.09 | 30.2 ± 1.17 | **0.93 ± 0.04** |
| + BlurGuard | **0.25 ± 0.07** | **0.81 ± 0.09** | **30.1 ± 2.19** | **101.84** | **0.35 ± 0.09** | **0.70 ± 0.09** | **28.2 ± 2.03** | **0.93 ± 0.05** |
| Mist [48] | 0.35 ± 0.12 | 0.72 ± 0.10 | 28.6 ± 0.25 | 90.96 | 0.30 ± 0.07 | 0.71 ± 0.09 | 29.7 ± 0.86 | 0.93 ± 0.04 |
| + BlurGuard | **0.33 ± 0.11** | **0.86 ± 0.08** | **32.3 ± 2.21** | **137.91** | **0.42 ± 0.10** | **0.63 ± 0.07** | **28.8 ± 0.47** | **0.89 ± 0.06** |

Table 5: BlurGuard combined with other adversarial objectives, *viz.*, AdvDM and Mist, on ImageNet-Edit. We consider $\ell_\infty$ protections within $\epsilon = \frac{16}{255}$.

As shown in Table 2, BlurGuard consistently maintains strong protection across these variations, demonstrating robustness to practical editing conditions.

**Instruction-based editing**   We also evaluate BlurGuard on instruction-based editing scenarios, focusing on InstructPix2Pix [7]; a fine-tuned diffusion model specifically to follow natural-language editing instructions. We use MagicBrush [98] dataset for the evaluation. As shown in Table 1c, BlurGuard shows superior worst-case protection effectiveness across all proposed metrics, while maintaining high imperceptibility competitive to the best baseline.

**Naturalness *vs.* robustness**   Figure 4 compares how the worst-case gap (measured by LPIPS) varies with the naturalness of each protection method (also measured by LPIPS) on image-to-image generation. Here, we define the worst-case gap as the degradation in protection effectiveness under the strongest purification method. To this end, we construct a histogram of pairs of these two LPIPS values for each method, drawing samples by varying the adversarial budget $\epsilon$ from $\frac{1}{255}$ to $\frac{16}{255}$. The results reveal that, unlike other methods that suffer from significant performance degradation after purification, BlurGuard consistently maintains a nearly zero performance gap across all LPIPS levels.

**Black-box transfer**   We further evaluate BlurGuard on black-box transfer. Specifically, we test image protections optimized from SD-v1.4 to other models; here we consider SD-v2.1. Results in Table 4 show that BlurGuard maintains its effectiveness in black-box setup as well, suggesting that adversarial noise in controlled frequency regime generalizes more effectively across models. Qualitative results can be found in Figure 13 in Appendix. We also test SD-v3.5, FLUX.1-dev [43], and a GAN-based SimSwap [14] as additional target models, as reported in Appendix D.2.

**BlurGuard with other adversarial objectives**   To demonstrate that BlurGuard operates in a simple plug-and-play manner, we test its compatibility with other adversarial objectives: AdvDM [49], which directly maximizes the denoising loss, and Mist [48], which jointly attacks the encoder and denoiser. As shown in Table 5, BlurGuard consistently enhances both naturalness and worst-case effectiveness across these objectives, demonstrating its broad applicability to diverse protection frameworks.

## 5.2 Ablation study

We perform an ablation study to validate the effectiveness of the individual components in BlurGuard. Throughout this study, we consider image-to-image generation tasks using SD-v1.4 with $\epsilon = \frac{16}{255}$. Additional ablation studies can be found in Appendix D.4.

| | Naturalness | | Worst-case Effect. | |
|---|---|---|---|---|
| | SSIM ↑ | PSNR ↑ | LPIPS ↑ | FID ↑ |
| **BlurGuard** | $0.93 \pm 0.09$ | $31.1 \pm 2.29$ | $0.32 \pm 0.09$ | 107.88 |
| w/o **Adaptive blurring** | | | | |
| $\quad \sigma = 0.0$ (const.) | $0.55 \pm 0.14$ | $32.9 \pm 2.49$ | $0.19 \pm 0.07$ | 65.94 |
| $\quad \sigma = 0.5$ (const.) | $0.59 \pm 0.13$ | $31.9 \pm 1.92$ | $0.22 \pm 0.10$ | 79.66 |
| $\quad \sigma = 1.0$ (const.) | $0.77 \pm 0.17$ | $31.6 \pm 1.44$ | $0.31 \pm 0.14$ | 94.00 |
| w/o **Per-region masks** | $0.94 \pm 0.09$ | $31.1 \pm 2.51$ | $0.29 \pm 0.09$ | 96.57 |
| w/o **Spectrum reg.** ($\lambda = 0.0$) | | | | |
| $\quad$ learned $\sigma$ | $0.69 \pm 0.12$ | $31.1 \pm 1.69$ | $0.40 \pm 0.11$ | 113.97 |
| $\quad$ learned $\sigma, \epsilon = \frac{4}{255}$ | $0.91 \pm 0.06$ | $32.0 \pm 1.13$ | $0.19 \pm 0.04$ | 28.57 |
| $\quad \sigma = 0.0$ | $0.70 \pm 0.11$ | $28.2 \pm 0.32$ | $0.27 \pm 0.07$ | 92.21 |

Table 6: Ablation study on BlurGuard. By default, we consider the $\ell_\infty$ constraint within $\epsilon = \frac{16}{255}$.

| Method | Inference Time (s) |
|---|---|
| PhotoGuard | $8.836 \pm 0.005$ |
| Mist | $31.267 \pm 0.044$ |
| AdvDM | $31.103 \pm 0.008$ |
| SDS | $12.493 \pm 0.016$ |
| **BlurGuard** | $\mathbf{31.071 \pm 0.012}$ |
| $\quad$– Semantic masking | $3.019 \pm 0.001$ |
| $\quad$– Adaptive blurring | $5.367 \pm 0.002$ |
| $\quad$– PGD updates | $22.685 \pm 0.010$ |

Table 7: Comparison of inference time (in seconds) measured on a single NVIDIA A100 (80GB) instance.

**Adaptive blurring**  To validate the effectiveness of our learnable Gaussian blur scheme, we compare BlurGuard with ablations where the blur intensity $\boldsymbol{\sigma}$ are all fixed as constant throughout optimizing $\boldsymbol{\delta}$. Specifically, we test $\boldsymbol{\sigma} = 0, 0.5$, and $1.0$. The results in Table 6 ("w/o Adaptive blurring") confirm that our adaptive $\sigma$ provides more effective protection by better aligning the frequency bands of the perturbation with the image. The significantly low SSIMs observed from the fixed-$\sigma$ configurations, despite their high PSNR scores, indicate that constant blurring often degrade image likelihood even when the perturbation magnitude is low. For example, fixed blurring may fail to account for textural characteristics of given image, leading to both unnatural and weak protection.

**Semantic-aware masks**  Next, we evaluate how our per-region blurring mechanism enhances BlurGuard, compared to its ablation ("w/o Per-region masks") that learns a single $\sigma$ for the entire image. Table 6 shows that the per-region approach provides higher protection effectiveness in worst-case scenarios, while being competitive in naturalness. This indicates that robust protection benefits from varying blurring intensities for different image regions depending on their frequency spectrum.

**Power spectrum regularization**  We test two ablations for our proposed power spectrum regularization: (a) $\lambda = 0$ while learning $\boldsymbol{\sigma}$ during the first training stage, and (b) those even without $\boldsymbol{\sigma}$, which is essentially equivalent to PhotoGuard (1) in this experiment. Overall, we observe that spectrum regularization plays a crucial role in preserving naturalness; although the adaptive blurring scheme already enhances worst-case effectiveness, it compromises naturalness without the regularization as the adversarial objective $L_{\text{adv}}$ in (10) alters the frequency spectrum. For example, the "learned $\sigma, \epsilon = \frac{4}{255}$" ablation yields far weaker worst-case effectiveness than BlurGuard while it maintains similar imperceptibility metrics. These results demonstrate that the spectrum regularization we propose induces a more effective protection without sacrificing visual imperceptibility.

**Computational overhead**  We report the computational overhead caused by each component of BlurGuard by measuring per-sample inference time (on a single NVIDIA A100 80GB GPU instance) as shown in Table 7. Overall, we observe that BlurGuard introduces overhead no greater than existing methods such as Mist and AdvDM, with the proposed adaptive blurring and semantic-aware masking pipeline accounting for only a small portion of the total time.

# 6   Conclusion

We introduce BlurGuard, a simple-yet-effective framework designed to protect images from unauthorized editing via generative models. BlurGuard tackles the current challenge of protection methods against purification by crafting adversarial noise that remains closer to in-distribution. We conduct extensive experiments to validate the effectiveness of BlurGuard, even introducing an "oracle" attacker equipped with the strongest purification among diverse options. The results consistently show that BlurGuard enhances robustness against existing purification methods, highlighting the potential of adversarial examples with naturalness as a promising direction for future research.

## Acknowledgements

This work was supported by the Institute of Information & Communications Technology Planning & Evaluation (IITP) grants funded by the Korea government (MSIT) (No. RS-2019-II190079, Artificial Intelligence Graduate School Program (Korea University); No. IITP-2025-RS-2025-02304828, Artificial Intelligence Star Fellowship Support Program to Nurture the Best Talents; No. IITP-2025-RS-2024-00436857, Information Technology Research Center (ITRC)), the National Research Foundation of Korea (NRF) grant funded by the Korea government (MSIT) (No. RS-2025-23523603), and the Korea Creative Content Agency grant funded by the Ministry of Culture, Sports and Tourism (No. RS-2025-00345025).

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

# A   Discussions

## A.1   Additional related work

**Safety concerns in generative AI**   Aside from the image protection technique that we focus on in this paper, generative AI has introduced various new safety concerns and there has been increasing attention to address them from a technical perspective. For instance, a line of works focuses on *deepfake detection* [6, 41, 47, 50, 61], *i.e.*, developing methods to detect machine-generated ("fake") content within real-world data. Another possible approach is *watermarking* [20, 24, 54, 60, 93, 100, 101], which aims to embed a unique identifier into generated images, enabling the identification of the images or their creators. Focusing more on the modeling perspective, a series of works has aimed to fine-tune models to remove specific concepts [27, 28, 32, 42, 97], *e.g.*, nudity or certain image styles, ensuring that these concepts are excluded from generated content. Other works have focused on fairness and unbiased generation [19, 25, 82, 86], in an attempt to mitigate biases in generative models that may generate harmful stereotypes.

**Adversarial examples for good**   Adversarial examples [5, 12, 87] were initially introduced to demonstrate that carefully crafted noise, imperceptible to human perception, can significantly change the outputs of machine learning models, causing them to make mistakes. Recent studies, however, have explored the potential of leveraging these adversarial examples for beneficial purposes, including robustifying classification models in imbalanced learning scenarios by generating synthetic minority class samples [39], enhancing model interpretability through counterfactual explanations, which illustrate minimal input changes needed to alter a model's prediction [36]. Notably, after the emergence of text-to-image diffusion models which can be easily misused for malicious purposes including *deepfake*, various studies [49, 77] have leveraged the adversarial examples against unauthorized editing facilitated by diffusion models.

## A.2   Limitation and broader impact

**Limitation**   A practical limitation of adversarial image protection is that, once an unprotected copy of an image is stolen and becomes publicly available, the perturbation can no longer effectively safeguard the image; an adversary can simply retrieve and edit the unprotected version. Consequently, future work could investigate model-level safeguards such as authorization systems that block unverified users from accessing the editing models, to prevent malicious editing at its source. Moreover, we note that the protective perturbation itself should not be regarded as a fundamental solution for manual editing scenarios by humans. While our approach may increase the cost of AI-based editing, it does not eliminate the threat posed by manual editing. Addressing such issues ultimately requires governance-level interventions beyond technical defenses.

**Broader impact**   Protecting images with adversarial perturbations offers a practical benefit to AI safety by impeding unauthorized, harmful AI-based image edits. However, by demonstrating the vulnerability of existing protections, although it has already been noted by Hönig et al. [34], we might inadvertently encourage malicious users to abuse this vulnerability for unauthorized image editing. We therefore hope this work motivates future research that further strengthens existing protection frameworks, rather than inspiring adversaries to exploit purification tools.

# B  Overall Procedure: BlurGuard

---

**Algorithm 1** BlurGuard

---

**Input:** source image $\mathbf{x}$; noise budget $\epsilon$; # steps $T_1$, $T_2$; step size $\gamma_1$, $\gamma_2$; loss coefficient $\lambda$
**Output:** protected image $\hat{\mathbf{x}}$

1: $\{M_r\}_{r=1}^R \leftarrow \text{SegmentAnything}(\mathbf{x})$      // Segment $\mathbf{x}$ into $R$ binary masks using SAM
2: Initialize $\boldsymbol{\delta} \leftarrow \mathbf{0}$
3: **for** each region $r \in \{1, \dots, R\}$ **do**
4:     Initialize $\omega_r \leftarrow 0$      // Parameterize the logarithm of $\boldsymbol{\sigma}$
5: **end for**
   {**Stage 1: Optimize $\omega$**}
6: Initialize $\boldsymbol{\delta}_0 \sim \mathcal{N}(\mathbf{0}, \mathbf{I})$      // Sample a Gaussian noise $\boldsymbol{\delta}_0$ for optimizing $\omega$
7: **for** each step $t = 1, \dots, T_1$ **do**
8:     $\boldsymbol{\sigma} \leftarrow \exp(\boldsymbol{\omega})$
9:     $\hat{\mathbf{x}} \leftarrow \mathbf{x} + \sum_{r=1}^R M_r \odot \mathcal{G}(\boldsymbol{\delta}_0, \sigma_r)$      // Aggregate noises with blurring to construct $\hat{\mathbf{x}}$
10:    $\boldsymbol{\omega} \leftarrow \text{Adam}(\boldsymbol{\omega}; L_{\text{freq}}(\hat{\mathbf{x}}, \mathbf{x}), \gamma_1)$      // Update $\omega$ via Adam
11: **end for**
   {**Stage 2: Optimize $\delta$**}
12: **for** each step $t = 1, \dots, T_2$ **do**
13:    $\boldsymbol{\sigma} \leftarrow \exp(\boldsymbol{\omega})$
14:    $\hat{\mathbf{x}} \leftarrow \mathbf{x} + \sum_{r=1}^R M_r \odot \mathcal{G}(\boldsymbol{\delta}, \sigma_r)$      // Adaptive blurring $\delta$ with the learned $\sigma$
15:    $L \leftarrow L_{\text{adv}}(\hat{\mathbf{x}}) + \lambda \cdot L_{\text{freq}}(\hat{\mathbf{x}}, \mathbf{x})$
16:    $\boldsymbol{\delta} \leftarrow \text{clip}(\boldsymbol{\delta} - \gamma_2 \cdot \frac{\nabla_{\boldsymbol{\delta}} L}{\|\nabla_{\boldsymbol{\delta}} L\|_2}; -\epsilon, \epsilon)$      // Update $\delta$ via PGD within noise budget $\epsilon$
17: **end for**
18: $\hat{\mathbf{x}} \leftarrow \mathbf{x} + \sum_{r=1}^R M_r \odot \mathcal{G}(\boldsymbol{\delta}, \sigma_r)$
19: **return** $\hat{\mathbf{x}}$

---

# C  Experimental Details

## C.1  Evaluation metrics

To quantitatively evaluate our methods, we selected multiple metrics to assess various aspects of image quality and the effectiveness of image protection. For image quality, we selected metrics that focus on perceptual naturalness as well as structural integrity and pixel-level distortions. To evaluate protection performance, we used metrics to compare perceptual, structural, and pixel-level differences between generated images from protected and unprotected source images. Additionally, we selected metrics to measure whether their embeddings and distributions have significantly diverged.

**LPIPS**  Learned Perceptual Image Patch Similarity (LPIPS) [99] measures the perceptual similarity between two images by comparing features extracted from a pre-trained neural network, such as VGG [83]. It focuses on high-level perceptual differences rather than pixel-wise accuracy.

**SSIM**  Structural Similarity Index (SSIM) evaluates the structural similarity between two images, such as an original image $\mathbf{X}$ and a transformed image $\mathbf{Y}$, by analyzing luminance ($l$), contrast ($c$), and structural patterns ($s$). It accounts for local pixel intensity patterns and combines these components into a single metric, defined as:

$$\text{SSIM}(\mathbf{X}, \mathbf{Y}) = [l(\mathbf{X}, \mathbf{Y})] \cdot [c(\mathbf{X}, \mathbf{Y})] \cdot [s(\mathbf{X}, \mathbf{Y})]. \tag{11}$$

**PSNR**  Peak Signal-to-Noise Ratio (PSNR) assesses the similarity between two images, such as an original image $\mathbf{x}$ and a noised image $\mathbf{y}$, by calculating the ratio between the maximum possible pixel intensity and the mean squared error (MSE). It provides a quantitative measure of how much the transformed image deviates from the original in terms of pixel-level fidelity, defined as:

$$\text{PSNR}(\mathbf{X}, \mathbf{Y}) = 20 \cdot \log_{10} \left( \frac{\text{MAX}(\mathbf{Y})}{\sqrt{\text{MSE}(\mathbf{X}, \mathbf{Y})}} \right). \tag{12}$$

**Image Alignment (IA) Score**   Image Alignment (IA) is a metric that measures the similarity between the embeddings of the source image and the protected image using cosine similarity. The embeddings are extracted using Contrastive Language-Image Pretraining (CLIP) [71], a pre-trained model that effectively captures high-level features from images by mapping them into an embedding space. The cosine similarity between the embeddings $\mathbf{e}_1$ and $\mathbf{e}_2$ of two images is defined as follows:

$$IA = \frac{\mathbf{e}_1 \cdot \mathbf{e}_2}{\|\mathbf{e}_1\|\|\mathbf{e}_2\|}. \tag{13}$$

**FID**   Fréchet Inception Distance (FID) evaluates the difference between two image sets: a set of source images $\mathbb{X}$ and a set of images $\mathbb{Y}$. It quantifies the difference in feature distributions between the two sets by extracting features with a pre-trained Inception network [88] and modeling their distributions as multivariate Gaussians, characterized by means $(\mu_{\mathbb{X}}, \mu_{\mathbb{Y}})$ and covariances $(\Sigma_{\mathbb{X}}, \Sigma_{\mathbb{Y}})$. The FID is calculated as:

$$FID = \|\mu_{\mathbb{X}} - \mu_{\mathbb{Y}}\|_2^2 + \text{Tr}(\Sigma_{\mathbb{X}} + \Sigma_{\mathbb{Y}} - 2(\Sigma_{\mathbb{X}}\Sigma_{\mathbb{Y}})^{1/2}). \tag{14}$$

## C.2   Implementation

**Baselines**   For image protection baselines, we consider 4 baseline image protection methods: AdvDM [49], Mist [48], PhotoGuard [77], and SDS [96]. We set the noise budget $\epsilon = 16/255$ and step size $\gamma = 2/255$ for each iteration of PGD attack. For PhotoGuard, we used their encoder attack with no target image as it is the simplest adversarial objective that can be easily integrated with our framework.

To ensure a comprehensive evaluation, we also include additional baselines for specific tasks. For the inpainting task, we compare our framework with DiffusionGuard [18], using the same noise budget and step size as in other baselines. For the textual inversion task, we include Glaze [80] as an additional baseline.

**Target models**   We use Stable Diffusion v1.4 [73] for image-to-image generation, textual inversion [26], and DreamBooth [74], while the inpainting model from Stable Diffusion v1.5 [73] is used for inpainting. For instruction-based editing, we use the pretrained InstructPix2Pix [7] model. For image-to-image generation and inpainting, we adopt the hyperparameter settings from PhotoGuard: 100 denoising steps, guidance scale of 7.5, and $\eta = 1$. For textual inversion, we follow the training and sampling procedures described in Hönig et al. [34]. For instruction-based editing, we apply the default hyperparameters provided in the InstructPix2Pix checkpoint on HuggingFace.[7]   For DreamBooth, we follow Anti-DreamBooth [90] for dataset sampling, training, and evaluation.

**BlurGuard**   For constructing our protection, we optimized the logarithm of the Gaussian blur intensities $\log \sigma_1, \ldots, \log \sigma_R$ using the Adam optimizer with a learning rate ($\gamma_1$) of $0.1$ during timesteps of $T_1 = 50$. After optimizing the blur intensities, we then optimize the adversarial perturbation with our l2-normalized PGD attack during timesteps of $T_2 = 100$ as same as other baseline protection methods. As the step size $\gamma_2$ for optimizing the perturbation $\delta$, we used $\gamma_2 = 20$ for image-to-image generation and textual inversion task, and $\gamma_2 = 50$ for inpainting.

## C.3   Datasets

**ImageNet-Edit**   Image-to-image generation offers high accessibility compared to techniques such as inpainting or textual inversion, which require additional masks or model training. However, this accessibility also increases the risk of misuse, necessitating countermeasures. Currently, there is a lack of datasets for developing these countermeasures, hindering consistent and meaningful evaluations in concurrent research. From this perspective, we curated a benchmark dataset for image-to-image generation task which is a subset of ImageNet [21] including 80 carefully selected images of animals and objects paired with editing prompts as illustrated in figure 5. We selected these images based on the following criteria: (a) images containing $40\% - 60\%$ background, making it easy to edit both the background and the main object, and (b) square images close to a 512×512 resolution, as the backbone model (Stable Diffusion v1.4) is trained on this resolution. The editing prompts were generated by using ChatGPT [65] to generate captions based on the original images and then make minor modifications to these captions, such as replacing certain nouns or adjectives.

---

[7]https://huggingface.co/timbrooks/instruct-pix2pix

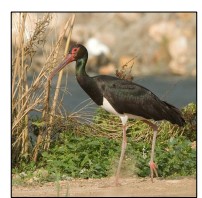 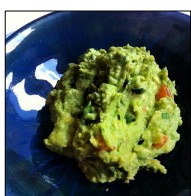 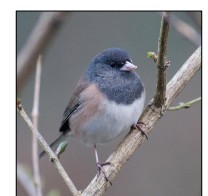 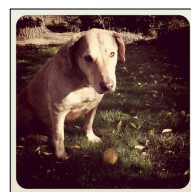

A white stork standing in a shallow wetland

A blue ceramic bowl filled with a salad

Junco bird perched on a snow-covered branch

Chesapeake Bay Retriever surrounded by blooming flowers in a garden.

Figure 5: Image-prompt pairs taken from ImageNet-Edit, a new benchmark dataset of 80 curated samples filtered from ImageNet, combined with ChatGPT for labeling, to evaluate image protection method in image-to-image generation task.

**Helen** For the inpainting task, we use the Helen dataset [44], following the pre-processing done by Cao et al. [10]. Specifically, they curated this dataset by sampling 80 images with the smallest face-to-image ratio to select images that are easy to modify. However, we found that the exact prompt set (used for editing) for this data is not publicly available; therefore we sample another set of prompts from the prompt set used by Choi et al. [18].

**WikiArt** We use the WikiArt dataset [89] for the textual inversion task. Each of the five artists (Albrecht Dürer, Edvard Munch, Alphonse Mucha, Edward Hopper, and Anna Ostroumova-Lebedeva) was selected based on [34], with 18 images assigned to each artist.

**MagicBrush** For the instruction-based editing task with InstructPix2Pix [7], we use the MagicBrush [98] dataset, which consists of images and corresponding instructions that can alter some aspects of the images. We randomly sampled 80 images and instructions from the test set of MagicBrush for our experiments.

**VGGFace2** For the DreamBooth [74] tasks in Appendix D.1, we use the VGGFace2 [11] dataset, which is a large-scale face recognition dataset. Similar to the WikiArt dataset for the textual inversion task. For evaluation, we sample 50 identities in VGGFace2, following the experimental settings of Anti-DreamBooth [90].

### C.4 Compute resources

We used a single NVIDIA A100 80GB GPU to run most of the experiments conducted; including processes of image protection, purification, and editing images. For fine-tuning Stable Diffusion models as done for the textual inversion experiments, we used a single NVIDIA RTX 3090 GPU, and a single NVIDIA RTX 3070 8GB to run Glaze.

### C.5 Purification methods

We provide an overview of the seven noise purification methods considered in our experiments. These methods include not only those specifically trained for adversarial purification but also general image processing techniques that essentially work as a noise purifier.

**JPEG compression** JPEG compression [91] is an image compression standard for enhancing the efficiency of storing images. While this compression process utilizes Discrete Cosine Transform (DCT) quantization steps, some high-frequency details of an image can be suppressed by this compression.

**Gaussian noise** Adding Gaussian noise into an image itself can be a noise purification tool, but with image upscaling [62], Hönig et al. [34] claims that one can effectively purify the adversarial noise without degradation in image quality. We also consider Gaussian noise with upscaling as one of our purification tools.

**Image upscaling**  Upscaling is a process that increases the resolution of an image by interpolating additional pixels. It can be used as a noise purification because resampling smooths high-frequency components, making it an effective tool for mitigating adversarial noise. An effective approach is to apply upscaling after performing JPEG compression or adding Gaussian noise, as these methods help suppress or disrupt adversarial patterns before the resampling step. Following this principle, we adopt both upscaling strategies as part of our purification methods.

**Impress**  IMPRESS [10] removes noise from protected images to ensure that they can reconstruct themselves when input into the diffusion model. Since protected images are visually very similar to the original images, the purified images are refined to retain similar visual characteristics to the protected images.

**DiffPure**  DiffPure [63] utilizes SDEdit [58], a purification method based on stochastic differential equations (SDE). DiffPure employs diffusion models for adversarial purification, effectively removing adversarial perturbations and recovering the original image. This process involves adding an appropriate level of noise during the forward diffusion step to neutralize adversarial patterns, followed by a reverse diffusion step to reconstruct a clean image.

**GrIDPure**  GrIDPure [102] is an extension of DiffPure [63]. It utilizes a grid-based slicing and merging process by dividing high-resolution images into overlapping grids, purifying each grid using the diffusion model, and then merging them back together.

**PDM-Pure**  PDM-Pure [94] is an adversarial purification method leveraging Pixel-Space Diffusion Models (PDM). It builds on SDEdit[58] by adding a small amount of noise to protected images and then performing a denoising process to remove protective patterns and restore the image closer to its original form.

# D  Additional Experiments

## D.1  More editing scenarios

| Method | Naturalness | | | Worst-case Effectiveness | | | | |
|---|---|---|---|---|---|---|---|---|
| | LPIPS ↓ | SSIM ↑ | PSNR ↑ | FID ↑ | LPIPS ↑ | SSIM ↓ | PSNR ↓ | IA ↓ |
| PhotoGuard [77] | $0.25 \pm 0.12$ | $0.77 \pm 0.08$ | $28.1 \pm 0.23$ | $177.18$ | $0.58 \pm 0.08$ | $0.25 \pm 0.10$ | $28.4 \pm 0.44$ | $0.89 \pm 0.05$ |
| AdvDM [49] | $0.26 \pm 0.11$ | $0.80 \pm 0.06$ | $28.8 \pm 0.28$ | $176.74$ | $\mathbf{0.61 \pm 0.09}$ | $\mathbf{0.23 \pm 0.09}$ | $\underline{28.4 \pm 0.42}$ | $0.88 \pm 0.04$ |
| Mist [48] | $0.24 \pm 0.12$ | $0.78 \pm 0.07$ | $28.7 \pm 0.23$ | $\mathbf{194.00}$ | $\mathbf{0.61 \pm 0.08}$ | $\mathbf{0.23 \pm 0.09}$ | $\underline{28.4 \pm 0.43}$ | $\underline{0.88 \pm 0.04}$ |
| SDS [96] | $0.22 \pm 0.10$ | $0.80 \pm 0.06$ | $29.0 \pm 0.37$ | $176.89$ | $0.57 \pm 0.08$ | $0.26 \pm 0.10$ | $\underline{28.4 \pm 0.39}$ | $0.89 \pm 0.05$ |
| Glaze [80] | $\mathbf{0.14 \pm 0.09}$ | $\underline{0.90 \pm 0.04}$ | $\underline{31.2 \pm 0.42}$ | $169.80$ | $0.57 \pm 0.09$ | $0.25 \pm 0.09$ | $28.5 \pm 0.51$ | $0.90 \pm 0.04$ |
| **BlurGuard (Ours)** | $\underline{0.15 \pm 0.08}$ | $\mathbf{0.97 \pm 0.03}$ | $\mathbf{32.1 \pm 3.06}$ | $\underline{181.89}$ | $\underline{0.59 \pm 0.08}$ | $0.25 \pm 0.10$ | $\mathbf{28.3 \pm 0.46}$ | $\mathbf{0.87 \pm 0.05}$ |

Table 8: Results on textual inversion with Stable Diffusion v1.4, under $\ell_\infty$ protection within $\epsilon = \frac{16}{255}$. *Naturalness* indicates how imperceptible the protection is. *Worst-case effectiveness* is measured after applying 7 different noise purification methods to the protected image, representing the robustness of each protection method. The best is highlighted in **bold**, while the second-best is underlined.

**Textual inversion**  Textual inversion [26] enables users to associate new visual concepts or styles into unique tokens via prompt tuning. However, it risks replicating copyrighted artwork and threatening intellectual property rights. We employed the WikiArt dataset [89], preprocessed as described in [34] to evaluate the performance of each protection methods. We fine-tune a text-to-image diffusion model on an image set $\mathbb{X}$ representing style using training steps = 2000, batch size = 4, and learning rate = $5 \times 10^{-6}$, following [34]. For each image $\mathbf{x}$, we generate a corresponding prompt $P_\mathbf{x} = C_\mathbf{x} +$ "by artist", where the image caption $C_\mathbf{x}$ is obtained using the BLIP-2 model [46]. The fine-tuned model generates images with a resolution of $768 \times 768$ using the DPM-Solver++ (2M) Karras scheduler [38, 53] over 50 sampling steps.

The results shows that BlurGuard surpassing most baselines in worst-case protection performance while maintaining superior image quality. Notably, BlurGuard achieves an LPIPS imperceptibility comparable to Glaze [80], which is explicitly optimized for minimizing LPIPS. Considering the inherent trade-off between imperceptibility and protection effectiveness, this substantially high imperceptibility suggests the potential for higher protection effectiveness while maintaining a similar level of imperceptibility to the baselines when a larger perturbation budget $\epsilon$ is used. This strong performance stems from our per-region blurring scheme, which adaptively incorporates textural information from images with diverse patterns. Given that the artworks used in textual inversion tasks often contain rich textures and vibrant colors, as shown in Figure 15, the high imperceptibility achieved by our method in these tasks is particularly meaningful.

**DreamBooth**  We also compared BlurGuard with other protection baselines on DreamBooth, which fine-tunes a diffusion model so that the generated images depict a user-specified concept. We used VGGFace2 [11] dataset to fine-tune Stable Diffusion v1.4. For the baselines, we employed Anti-DreamBooth [90] and High-Frequency Anti-DreamBooth [64], which are specially designed frameworks to protect images against DreamBooth. Table 9 demonstrates that BlurGuard surpasses other protection methods including High-Frequency Anti-Dreambooth, which selectively applies large amount of perturbation only in high-frequency areas of the image. Moreover, the results show that BlurGuard can also be combined with other protection methods as a plug-and-play manner, supporting its high applicability.

| Method | LPIPS ↑ | FID ↑ | SSIM ↓ | PSNR ↓ | IA ↓ |
|---|---|---|---|---|---|
| Anti-DreamBooth [90] | $\mathbf{0.12 \pm 0.02}$ | $127.93$ | $\underline{0.51 \pm 0.11}$ | $28.5 \pm 0.47$ | $0.88 \pm 0.06$ |
| High-Frequency Anti-DreamBooth [64] | $\mathbf{0.12 \pm 0.05}$ | $\underline{138.45}$ | $\underline{0.53 \pm 0.10}$ | $28.6 \pm 0.43$ | $0.89 \pm 0.05$ |
| **BlurGuard + Anti-DreamBooth** | $\mathbf{0.12 \pm 0.10}$ | $\mathbf{145.75}$ | $\mathbf{0.49 \pm 0.10}$ | $\mathbf{28.3 \pm 0.35}$ | $\mathbf{0.86 \pm 0.06}$ |

Table 9: Worst-case effectiveness of different image protection methods on Dreambooth, under $\ell_\infty$ protection within $\epsilon = \frac{16}{255}$. *Worst-case effectiveness* is measured after applying 7 different noise-purification methods. The best value is in **bold**, the second-best is underlined.

## D.2  Black-box transfer

| IN-Edit ($\varepsilon = \frac{16}{255}$) | (a) SD-v1.4 → SD-v3.5 | | | | | (b) SD-v1.4 → FLUX.1-dev | | | | |
|---|---|---|---|---|---|---|---|---|---|---|
| Method | FID ↑ | LPIPS ↑ | SSIM ↓ | PSNR ↓ | IA ↓ | FID ↑ | LPIPS ↑ | SSIM ↓ | PSNR ↓ | IA ↓ |
| PhotoGuard [77] | 112.58 | 0.40 ± 0.09 | 0.48 ± 0.18 | 29.1 ± 0.85 | 0.90 ± 0.04 | 72.60 | **0.26 ± 0.08** | 0.80 ± 0.07 | 30.7 ± 0.95 | 0.95 ± 0.03 |
| AdvDM [49] | 98.25 | 0.34 ± 0.10 | 0.71 ± 0.09 | 29.9 ± 1.04 | 0.92 ± 0.04 | 71.74 | **0.26 ± 0.08** | **0.78 ± 0.06** | 30.6 ± 1.06 | 0.94 ± 0.04 |
| Mist [48] | **120.17** | 0.43 ± 0.08 | 0.47 ± 0.19 | 28.8 ± 0.83 | 0.89 ± 0.05 | 63.59 | 0.21 ± 0.06 | 0.80 ± 0.06 | 30.3 ± 0.87 | 0.95 ± 0.03 |
| SDS [96] | 102.12 | 0.41 ± 0.09 | 0.52 ± 0.14 | 29.4 ± 0.88 | 0.89 ± 0.05 | 58.59 | 0.19 ± 0.06 | 0.81 ± 0.07 | 31.0 ± 1.37 | 0.96 ± 0.03 |
| **BlurGuard (Ours)** | 117.71 | **0.46 ± 0.08** | **0.46 ± 0.19** | **28.4 ± 0.54** | **0.88 ± 0.05** | 74.34 | **0.26 ± 0.09** | 0.79 ± 0.09 | **29.0 ± 0.69** | **0.93 ± 0.04** |

Table 10: Comparison of worst-case effectiveness on ImageNet- Edit for black-box transfer. We consider two additional scenarios **(a)** SD-v1.4 → SD-v3.5 and **(b)** SD-v1.4 → FLUX.1-dev. The best is highlighted in **bold**, while the second-best is underlined.

| SD-v1.4 → SimSwap | Naturalness | | | Worst-case Effectiveness | | | | |
|---|---|---|---|---|---|---|---|---|
| Method | LPIPS ↓ | SSIM ↑ | PSNR ↑ | FID ↑ | LPIPS ↑ | SSIM ↓ | PSNR ↓ | IA ↓ |
| PhotoGuard [77] | 0.04 ± 0.02 | 0.96 ± 0.02 | 38.2 ± 2.53 | 18.34 | **0.03 ± 0.02** | 0.97 ± 0.02 | 40.0 ± 2.67 | 0.99 ± 0.00 |
| AdvDM [49] | 0.04 ± 0.03 | 0.97 ± 0.01 | **40.7 ± 2.65** | 17.16 | **0.03 ± 0.02** | 0.97 ± 0.01 | 40.4 ± 2.43 | 0.99 ± 0.00 |
| Mist [48] | **0.02 ± 0.01** | 0.98 ± 0.01 | 40.5 ± 2.60 | 14.10 | 0.02 ± 0.02 | **0.98 ± 0.01** | 41.0 ± 2.43 | 0.99 ± 0.00 |
| SDS [96] | 0.06 ± 0.03 | 0.96 ± 0.02 | 38.9 ± 2.55 | 25.11 | **0.03 ± 0.02** | 0.97 ± 0.02 | 39.3 ± 2.44 | 0.99 ± 0.01 |
| DiffusionGuard [18] | 0.04 ± 0.02 | 0.97 ± 0.01 | 39.6 ± 2.49 | 17.39 | 0.02 ± 0.02 | 0.97 ± 0.01 | 40.2 ± 2.33 | 0.99 ± 0.00 |
| **BlurGuard (Ours)** | 0.03 ± 0.02 | **0.99 ± 0.01** | 38.7 ± 2.22 | **31.54** | 0.03 ± 0.01 | **0.98 ± 0.01** | **38.4 ± 2.37** | **0.97 ± 0.02** |

Table 11: Black-box transfer on InpaintGuardBench for inpainting (SD-v1.4 → SimSwap), under $\ell_\infty$ protection within $\epsilon = \frac{16}{255}$. The best is highlighted in **bold**, while the second-best is underlined.

**Transferability across diffusion models**  We further examine the black-box transferability of BlurGuard by evaluating its performance when adversarial protections from SD-v1.4 are transferred to newer diffusion models, including SD-v3.5 and FLUX.1-dev [43]. Note that these target models use the MMDiT architecture with flow-matching, which substantially differ from SD-v1.4. As shown in Table 10, BlurGuard consistently improves performance over PhotoGuard across all metrics, demonstrating its clear advantage in enhancing transferability. Moreover, BlurGuard maintains strong and competitive protection performance across all target models, highlighting the general applicability of our framework to diverse generative architectures. These results show that BlurGuard offers a broadly transferable solution for adversarial image protection in diffusion-based models.

**Transferability to GAN**  To examine the cross-framework transferability of BlurGuard, we conduct a black-box transfer experiment from SD-v1.4 to SimSwap [14], a GAN-based method for deepfake generation. We perform the evaluation on the human portrait subset of InpaintGuardBench [18]. As shown in Table 11, BlurGuard maintains strong performance in both naturalness and worst-case effectiveness, confirming that the protective perturbations generated by our framework remain effective even when transferred across fundamentally different generative frameworks from diffusion models to GANs. These results highlight the robustness and broad generalizability of BlurGuard beyond diffusion-based architectures.

## D.3  Evaluation with other diffusion models

| SD-v2.1 ($\varepsilon = \frac{16}{255}$) | Naturalness | | | Worst-case Effectiveness | | | | |
|---|---|---|---|---|---|---|---|---|
| Method | LPIPS ↓ | SSIM ↑ | PSNR ↑ | FID ↑ | LPIPS ↑ | SSIM ↓ | PSNR ↓ | IA ↓ |
| PhotoGuard [77] | $0.34 \pm 0.11$ | $0.70 \pm 0.11$ | $28.2 \pm 0.32$ | $\underline{145.29}$ | $0.36 \pm 0.07$ | $0.66 \pm 0.12$ | $29.4 \pm 0.85$ | $0.91 \pm 0.04$ |
| AdvDM [49] | $\underline{0.25 \pm 0.10}$ | $\underline{0.79 \pm 0.09}$ | $29.4 \pm 0.33$ | $140.76$ | $0.35 \pm 0.07$ | $0.67 \pm 0.12$ | $29.7 \pm 0.96$ | $0.91 \pm 0.04$ |
| Mist [48] | $0.31 \pm 0.12$ | $0.75 \pm 0.09$ | $28.9 \pm 0.25$ | $143.37$ | $\underline{0.38 \pm 0.07}$ | $0.66 \pm 0.12$ | $29.6 \pm 0.91$ | $\underline{0.90 \pm 0.04}$ |
| SDS [96] | $0.32 \pm 0.11$ | $\underline{0.79 \pm 0.08}$ | $\underline{29.8 \pm 0.57}$ | $136.36$ | $0.36 \pm 0.08$ | $0.67 \pm 0.12$ | $29.7 \pm 1.00$ | $0.91 \pm 0.04$ |
| **BlurGuard (Ours)** | $\mathbf{0.22 \pm 0.10}$ | $\mathbf{0.93 \pm 0.09}$ | $\mathbf{31.1 \pm 2.27}$ | $\mathbf{154.46}$ | $\mathbf{0.41 \pm 0.07}$ | $\mathbf{0.62 \pm 0.11}$ | $\mathbf{28.7 \pm 0.55}$ | $\mathbf{0.89 \pm 0.04}$ |

Table 12: Results on image-to-image generation with SD-v2.1, under $\ell_\infty$ protection within $\epsilon = \frac{16}{255}$. *Naturalness* indicates how imperceptible the protection is. *Worst-case effectiveness* is measured after applying 7 different noise purification methods to the protected image, representing the robustness of each protection method. The best is highlighted in **bold**, while the second-best is underlined.

| SD-v3.5 ($\varepsilon = \frac{16}{255}$) | Naturalness | | | Worst-case Effectiveness | | | | |
|---|---|---|---|---|---|---|---|---|
| Method | LPIPS ↓ | SSIM ↑ | PSNR ↑ | FID ↑ | LPIPS ↑ | SSIM ↓ | PSNR ↓ | IA ↓ |
| PhotoGuard [77] | $\mathbf{0.24 \pm 0.09}$ | $0.79 \pm 0.09$ | $\mathbf{31.5 \pm 1.30}$ | $\underline{121.58}$ | $0.45 \pm 0.08$ | $0.48 \pm 0.19$ | $28.6 \pm 0.64$ | $\mathbf{0.88 \pm 0.05}$ |
| AdvDM [49] | $0.28 \pm 0.10$ | $0.77 \pm 0.08$ | $29.5 \pm 0.26$ | $116.36$ | $0.42 \pm 0.08$ | $0.48 \pm 0.19$ | $29.0 \pm 0.94$ | $\underline{0.89 \pm 0.05}$ |
| Mist [48] | $\underline{0.25 \pm 0.09}$ | $0.80 \pm 0.09$ | $\underline{30.9 \pm 1.14}$ | $118.91$ | $0.45 \pm 0.08$ | $0.48 \pm 0.19$ | $28.6 \pm 0.64$ | $\mathbf{0.88 \pm 0.05}$ |
| SDS [96] | $\underline{0.25 \pm 0.11}$ | $\underline{0.86 \pm 0.08}$ | $29.5 \pm 0.32$ | $119.20$ | $\underline{0.46 \pm 0.08}$ | $\underline{0.47 \pm 0.19}$ | $\underline{28.5 \pm 0.56}$ | $\mathbf{0.88 \pm 0.05}$ |
| **BlurGuard (Ours)** | $\underline{0.25 \pm 0.10}$ | $\mathbf{0.88 \pm 0.13}$ | $30.4 \pm 1.97$ | $\mathbf{125.32}$ | $\mathbf{0.48 \pm 0.07}$ | $\mathbf{0.46 \pm 0.18}$ | $\mathbf{28.4 \pm 0.46}$ | $\mathbf{0.88 \pm 0.05}$ |

Table 13: Results on image-to-image generation with SD-v3.5, under $\ell_\infty$ protection within $\epsilon = \frac{16}{255}$. *Naturalness* indicates how imperceptible the protection is. *Worst-case effectiveness* is measured after applying 7 different noise purification methods to the protected image, representing the robustness of each protection method. The best is highlighted in **bold**, while the second-best is underlined.

| FLUX.1-dev ($\varepsilon = \frac{16}{255}$) | Naturalness | | | Worst-case Effectiveness | | | | |
|---|---|---|---|---|---|---|---|---|
| Method | LPIPS ↓ | SSIM ↑ | PSNR ↑ | FID ↑ | LPIPS ↑ | SSIM ↓ | PSNR ↓ | IA ↓ |
| PhotoGuard [77] | $\mathbf{0.21 \pm 0.10}$ | $0.86 \pm 0.07$ | $32.1 \pm 0.33$ | $80.22$ | $0.28 \pm 0.06$ | $\underline{0.68 \pm 0.11}$ | $29.5 \pm 0.31$ | $0.91 \pm 0.05$ |
| AdvDM [49] | $0.27 \pm 0.10$ | $0.85 \pm 0.07$ | $\underline{32.6 \pm 1.10}$ | $83.42$ | $0.28 \pm 0.06$ | $\underline{0.68 \pm 0.12}$ | $28.8 \pm 0.72$ | $\underline{0.90 \pm 0.05}$ |
| Mist [48] | $0.26 \pm 0.09$ | $0.87 \pm 0.08$ | $32.3 \pm 1.40$ | $\underline{84.10}$ | $\underline{0.29 \pm 0.07}$ | $\mathbf{0.67 \pm 0.12}$ | $29.7 \pm 0.60$ | $\underline{0.90 \pm 0.04}$ |
| SDS [96] | $0.24 \pm 0.10$ | $\underline{0.88 \pm 0.08}$ | $32.2 \pm 0.80$ | $81.65$ | $0.28 \pm 0.06$ | $\underline{0.68 \pm 0.11}$ | $\underline{28.6 \pm 0.48}$ | $\underline{0.90 \pm 0.05}$ |
| **BlurGuard (Ours)** | $\underline{0.23 \pm 0.11}$ | $\mathbf{0.89 \pm 0.07}$ | $\mathbf{33.4 \pm 2.90}$ | $\mathbf{85.21}$ | $\mathbf{0.30 \pm 0.10}$ | $\mathbf{0.67 \pm 0.13}$ | $\mathbf{28.4 \pm 0.33}$ | $\mathbf{0.89 \pm 0.05}$ |

Table 14:  Results on image-to-image generation with FLUX.1-dev, under $\ell_\infty$ protection within $\epsilon = \frac{16}{255}$. *Naturalness* indicates how imperceptible the protection is. *Worst-case effectiveness* is measured after applying 7 different noise purification methods to the protected image, representing the robustness of each protection method. The best is highlighted in **bold**, while the second-best is underlined.

We further investigate the generalizability of BlurGuard across more recent diffusion models by constructing protections against Stable Diffusion v2.1 (SD-v2.1), SD-v3.5, and FLUX.1-dev. In particular, SD-v3.5 and FLUX.1-dev adopt the MMDiT architecture with flow-matching training, representing a substantially different design paradigm from SD-v1.4 used in most of our experiments. As summarized in Tables 12, 13, and 14, BlurGuard consistently surpasses all baseline protection methods across nearly all metrics, demonstrating its robustness and scalability to newer, more advanced diffusion backbones. Figure 6 also supports these findings qualitatively.

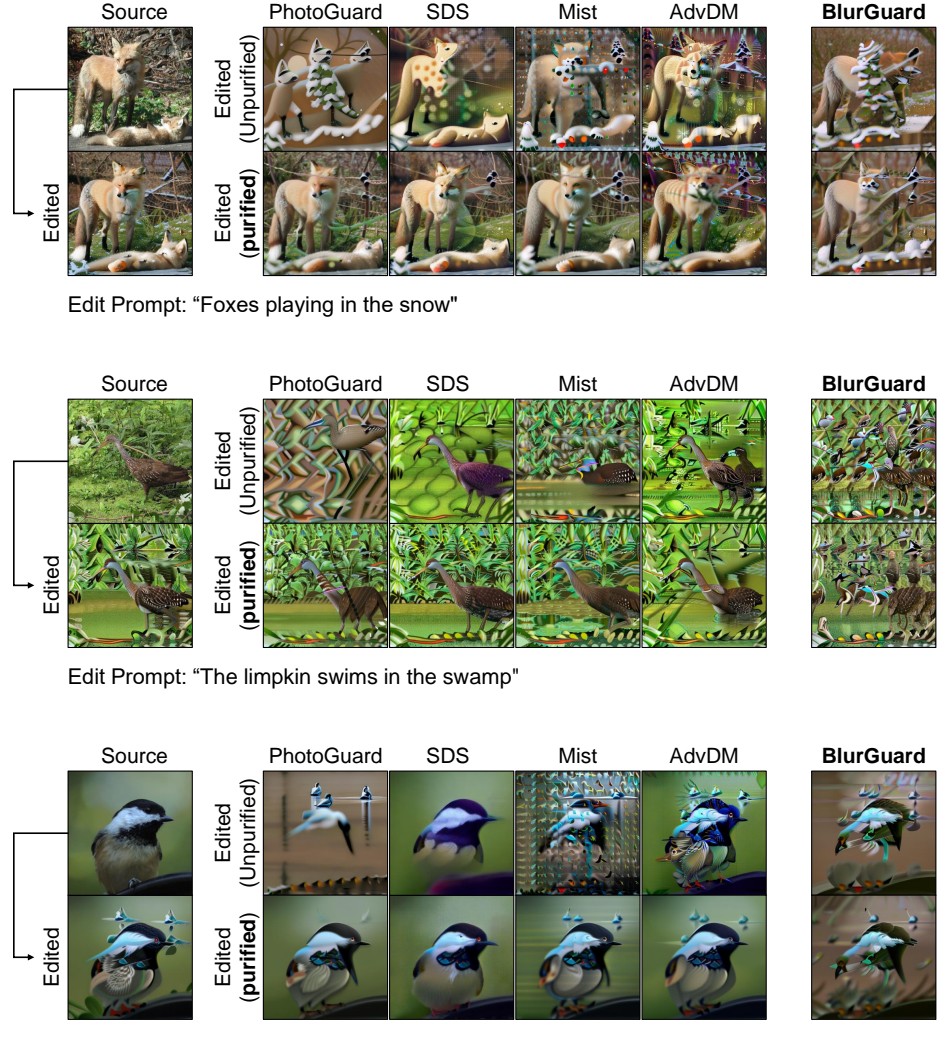

Figure 6: Qualitative comparison on image-to-image generation using Stable Diffusion v2.1.

## D.4 Ablation study

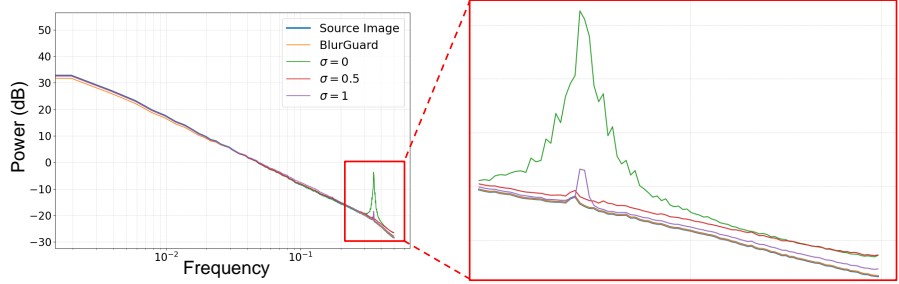

Figure 7: Comparison of the Radially-Averaged Power Spectral Density (RAPSD) of image protections using constant Gaussian blur intensity $\sigma$. The deviation in high-frequency bands (right side of the plot) often corresponds to the vulnerability against noise purification.

**On the constant $\sigma$ ablations** We have shown that adaptive blurring is effective both for the naturalness and robustness of image protection (see Section 5.2). In Figure 7, we further show how the different blur intensities $\sigma$ affect the frequency spectrum of protected images by showing their RAPSD. Unlike the original image that shows a linear RAPSD trend, protected images with non-blurred perturbation ($\sigma = 0$) or perturbation with fixed blur intensity ($\sigma = 0.5, 1$) show large deviations in high-frequency levels. This deviation in high-frequency bands can directly harm the robustness of protection. These observations support the low protection performances shown by constant $\sigma$ ablations ($\sigma = 0.0, 0.5, 1.0$) reported in Table 6.

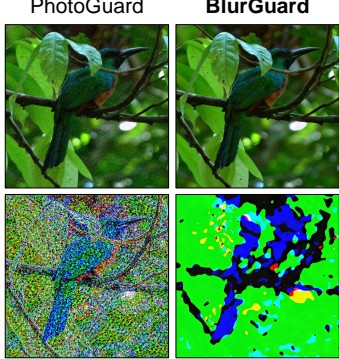

PhotoGuard **BlurGuard**

Figure 8: Visualization of the added noise from PhotoGuard and BlurGuard (ours).

**Visualization of perturbation** We present a qualitative comparison of actual perturbations between PhotoGuard and our proposed BlurGuard in Figure 8; overall, we confirm that PhotoGuard tends to generate high-frequency perturbations uniformly across the image, whereas BlurGuard applies adaptive blurring per region, producing perturbations that are both more natural and harder to purify.

**Lower perturbation budget** We report the results on image-to-image task under $\ell_\infty$ protection within $\epsilon = \frac{8}{255}$, which can be more practical when the perturbation is required to be merely perceptible. We show that BlurGuard also demonstrates its effectiveness both in quantitative results in Table 15. While the PSNR naturalness of BlurGuard may appear slightly lower, this reflects a deliberate design choice rather than higher perceptibility. PSNR focuses on pixel-wise fidelity rather than perceptual similarity and often penalizes imperceptible local deviations. In contrast, BlurGuard intentionally allocates perturbations within perceptually insensitive regions through region-wise frequency adaptation, which can lower PSNR while preserving imperceptibility, as qualitatively illustrated in Figure 11.

| IN-Edit ($\epsilon = \frac{8}{255}$) | Naturalness | | | Worst-case Effectiveness | | | | |
|---|---|---|---|---|---|---|---|---|
| Method | **LPIPS ↓** | **SSIM ↑** | **PSNR ↑** | **FID ↑** | **LPIPS ↑** | **SSIM ↓** | **PSNR ↓** | **IA ↓** |
| PhotoGuard [77] | **0.06 ± 0.04** | 0.89 ± 0.05 | 32.4 ± 0.38 | 57.31 | 0.14 ± 0.06 | 0.83 ± 0.06 | 31.6 ± 1.25 | 0.97 ± 0.03 |
| AdvDM [49] | 0.14 ± 0.07 | 0.88 ± 0.05 | 32.6 ± 0.50 | 83.31 | **0.24 ± 0.06** | **0.76 ± 0.08** | 30.6 ± 1.30 | 0.95 ± 0.03 |
| Mist [48] | 0.12 ± 0.05 | 0.86 ± 0.06 | 32.0 ± 0.45 | 73.21 | 0.21 ± 0.07 | 0.77 ± 0.08 | 30.6 ± 1.18 | 0.95 ± 0.03 |
| SDS [96] | 0.09 ± 0.03 | 0.87 ± 0.06 | **32.9 ± 0.54** | 71.28 | 0.19 ± 0.06 | 0.79 ± 0.08 | 30.9 ± 1.28 | 0.96 ± 0.03 |
| **BlurGuard (Ours)** | 0.08 ± 0.05 | **0.95 ± 0.07** | 30.7 ± 0.46 | **84.38** | 0.23 ± 0.08 | 0.77 ± 0.08 | **29.9 ± 0.89** | **0.94 ± 0.04** |

Table 15: Results on image-to-image generation with SD-v1.4, under $\ell_\infty$ protection within $\epsilon = \frac{8}{255}$. *Naturalness* indicates how imperceptible the protection is. *Worst-case effectiveness* is measured after applying 7 different noise purification methods to the protected image, representing the robustness of each protection method. The best is highlighted in **bold**, while the second-best is underlined.

**Lower image resolutions**  To reflect real-world scenarios where we often encounter variable image resolutions, we additionally evaluate BlurGuard at lower image resolutions, *viz.*, 128×128 and 256×256. For the evaluation, we down-sample the images in ImageNet-Edit dataset to each target resolution; the results are summarized in Table 16a and Table 16b. Interestingly, BlurGuard stands out as the only approach that could effectively preserve its effectiveness, whereas all the other considered baselines commonly suffer from significant degradation in worst-case effectiveness during the resampling procedure in handling different resolutions. These results underscore the utility of BlurGuard in a practical setup when input images differ from the 512×512 resolution on which the Stable Diffusion model is primarily trained.

| IN-Edit ($\epsilon = \frac{16}{255}$) | Naturalness | | | Worst-case Effectiveness | | | | |
|---|---|---|---|---|---|---|---|---|
| Method | LPIPS ↓ | SSIM ↑ | PSNR ↑ | FID ↑ | LPIPS ↑ | SSIM ↓ | PSNR ↓ | IA ↓ |
| PhotoGuard [77] | **0.09 ± 0.06** | 0.81 ± 0.09 | 29.8 ± 0.57 | 70.88 | 0.07 ± 0.04 | 0.92 ± 0.03 | 32.2 ± 1.41 | 0.96 ± 0.03 |
| AdvDM [49] | **0.09 ± 0.05** | 0.85 ± 0.08 | 29.3 ± 0.27 | 82.57 | 0.09 ± 0.04 | 0.91 ± 0.04 | 31.5 ± 0.99 | 0.96 ± 0.02 |
| Mist [48] | 0.17 ± 0.07 | 0.78 ± 0.10 | 29.7 ± 0.62 | 118.35 | 0.12 ± 0.06 | 0.87 ± 0.05 | 30.3 ± 0.81 | 0.94 ± 0.03 |
| SDS [96] | 0.10 ± 0.04 | 0.83 ± 0.08 | 29.6 ± 0.35 | 82.06 | 0.08 ± 0.04 | 0.91 ± 0.03 | 31.9 ± 1.39 | 0.96 ± 0.02 |
| **BlurGuard (Ours)** | 0.10 ± 0.04 | **0.92 ± 0.08** | **32.5 ± 3.54** | **155.33** | **0.23 ± 0.08** | **0.84 ± 0.05** | **28.8 ± 0.56** | **0.90 ± 0.05** |

(a) Results on 128 × 128 resolution.

| IN-Edit ($\epsilon = \frac{16}{255}$) | Naturalness | | | Worst-case Effectiveness | | | | |
|---|---|---|---|---|---|---|---|---|
| Method | LPIPS ↓ | SSIM ↑ | PSNR ↑ | FID ↑ | LPIPS ↑ | SSIM ↓ | PSNR ↓ | IA ↓ |
| PhotoGuard [77] | 0.16 ± 0.08 | 0.75 ± 0.10 | 30.0 ± 0.60 | 73.80 | 0.13 ± 0.05 | 0.89 ± 0.04 | 31.8 ± 1.44 | 0.96 ± 0.03 |
| AdvDM [49] | 0.15 ± 0.07 | 0.81 ± 0.09 | 29.4 ± 0.27 | 76.72 | 0.14 ± 0.05 | 0.88 ± 0.04 | 31.4 ± 1.33 | 0.96 ± 0.02 |
| Mist [48] | 0.22 ± 0.08 | 0.75 ± 0.10 | 29.7 ± 0.45 | 96.34 | 0.16 ± 0.06 | 0.86 ± 0.05 | 30.6 ± 1.17 | 0.95 ± 0.03 |
| SDS [96] | 0.17 ± 0.05 | 0.78 ± 0.08 | 29.4 ± 0.47 | 93.78 | 0.14 ± 0.05 | 0.87 ± 0.05 | 31.3 ± 1.47 | 0.95 ± 0.03 |
| **BlurGuard (Ours)** | **0.13 ± 0.06** | **0.92 ± 0.09** | **31.8 ± 2.95** | **121.84** | **0.27 ± 0.09** | **0.81 ± 0.06** | **28.8 ± 0.57** | **0.92 ± 0.04** |

(b) Results on 256 × 256 resolution.

Table 16: Results of image-to-image generation on low-resolution inputs under $\ell_\infty$ protection with $\epsilon = \frac{16}{255}$. *Naturalness* indicates how imperceptible the protection is. *Worst-case effectiveness* is measured after applying 7 different noise purification methods to the protected image. The best is highlighted in **bold**, while the second-best is underlined.

**Adaptive blurring parameter**  Table 17 shows an ablation where we vary the SAM hyperparameter related to the number of masks that BlurGuard optimizes. Allowing more masks slightly lowers the naturalness but consistently improves worst-case effectiveness. This is likely because a higher number of masks enables a more flexible optimization in BlurGuard. However, removing adaptive blurring itself significantly degrades the protection, verifying its importance to BlurGuard.

| IN-Edit ($\epsilon = \frac{16}{255}$) | Naturalness | Worst-case Effect. | |
|---|---|---|---|
| **Method** | SSIM ↑ | FID ↑ | LPIPS ↑ |
| w/o Per-region masks. | 0.94 | 96.6 | 0.29 |
| Avg. # Masks = 10.3. | 0.91 | 108.5 | 0.34 |
| Avg. # Masks = 21.4. | 0.89 | 115.9 | 0.36 |
| Avg # Masks = 4.7 (Ours) | 0.93 | 107.9 | 0.32 |

Table 17: Additional ablation results on number of masks in adaptive blurring.

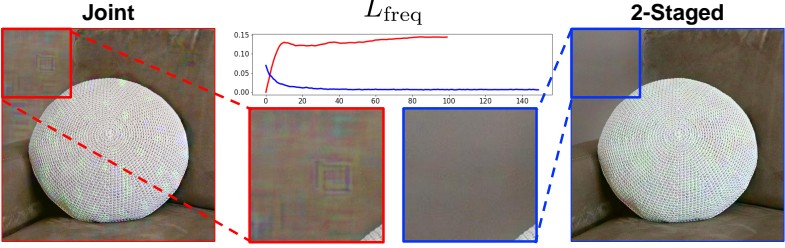

Figure 9: Ablation study on two-stage optimization of BlurGuard. Qualitative comparison between BlurGuard and its variant which jointly optimizes the blur intensities $\sigma$ and the adversarial perturbation $\delta$. We also report the trend of $L_{\text{freq}}$ in each setting.

**Two-stage optimization**  We also compare our two-stage optimization scheme with its ablation, *i.e.*, that jointly optimizes the two objectives $L_{\text{adv}}$ and $L_{\text{freq}}$. As observed in Figure 9, the two-stage optimization can help to stabilize the optimization of frequency loss $L_{\text{freq}}$. In qualitative manner, on the other hand, we observe that the joint optimization (left) often generates unnatural artifacts with patterns in their protection; this happens when the frequency loss $L_{\text{freq}}$ is minimized by tuning $\boldsymbol{\delta}$, rather than $\boldsymbol{\sigma}$, making Gaussian blur less effective.

**Detailed ablation on blurring design**  Note that BlurGuard applies adaptive blurring only to $\boldsymbol{\delta}$, not the entire image $\mathbf{x} + \boldsymbol{\delta}$. To verify this adaptive blurring design as an effective way to control its frequency band, we compare BlurGuard with two frequency adaptation variants. Specifically, we apply (i) *global blurring* to the entire protected image $\mathbf{x} + \boldsymbol{\delta}$, and (ii) *sharpening* to the perturbation $\boldsymbol{\delta}$ using unsharp masking [68]. As shown in Table 18, these variants underperform BlurGuard in both naturalness and worst-case effectiveness of protection. Global blurring moderately improves robustness but shows a notable drop in SSIM and LPIPS naturalness metrics compared to BlurGuard, since blurring the entire image removes high-frequency details and thus compromises perceptual quality. The adaptive sharpening scheme fails to provide effective protection after purification because sharpening amplifies high-frequency components that are easily suppressed by existing purification techniques such as JPEG compression.

| IN-Edit ($\varepsilon = \frac{16}{255}$) | Naturalness | | | Worst-case Effectiveness | | | | |
|---|---|---|---|---|---|---|---|---|
| Method | LPIPS ↓ | SSIM ↑ | PSNR ↑ | FID ↑ | LPIPS ↑ | SSIM ↓ | PSNR ↓ | IA ↓ |
| PhotoGuard [77] | $0.34 \pm 0.11$ | $0.70 \pm 0.11$ | $28.2 \pm 0.32$ | $92.21$ | $0.27 \pm 0.07$ | $0.73 \pm 0.10$ | $\underline{29.9 \pm 1.07}$ | $\underline{0.94 \pm 0.04}$ |
| $\mathbf{x} + \text{Blur}(\boldsymbol{\delta})$ | $\mathbf{0.21 \pm 0.10}$ | $\mathbf{0.93 \pm 0.09}$ | $\underline{31.1 \pm 2.29}$ | $\mathbf{107.88}$ | $\mathbf{0.32 \pm 0.09}$ | $\mathbf{0.70 \pm 0.09}$ | $\mathbf{28.8 \pm 0.42}$ | $\mathbf{0.92 \pm 0.05}$ |
| $\text{Blur}(\mathbf{x} + \boldsymbol{\delta})$ | $0.32 \pm 0.11$ | $0.79 \pm 0.08$ | $\mathbf{32.3 \pm 1.11}$ | $\underline{98.45}$ | $0.30 \pm 0.07$ | $0.71 \pm 0.10$ | $30.1 \pm 1.21$ | $0.94 \pm 0.04$ |
| $\text{Sharpen}(\boldsymbol{\delta})$ | $0.35 \pm 0.12$ | $0.65 \pm 0.10$ | $30.9 \pm 1.08$ | $82.72$ | $0.28 \pm 0.08$ | $0.72 \pm 0.10$ | $30.4 \pm 1.38$ | $0.94 \pm 0.04$ |

Table 18: Comparison of BlurGuard with additional blurring design ablations.

# E  Additional Qualitative Results

In this section, we report additional qualitative results of our proposed method, BlurGuard, to further support its effectiveness. In Figure 10, we report additional qualitative results on the black-box transfer experiments (Section 5.1). Figure 12, Figure 14, and Figure 15 compares BlurGuard with other baselines in image-to-image generation, inpainting, and textual inversion tasks, respectively. Figure 6 supplies qualitative results on the Stable Diffusion v2.1 model (Appendix D.3).

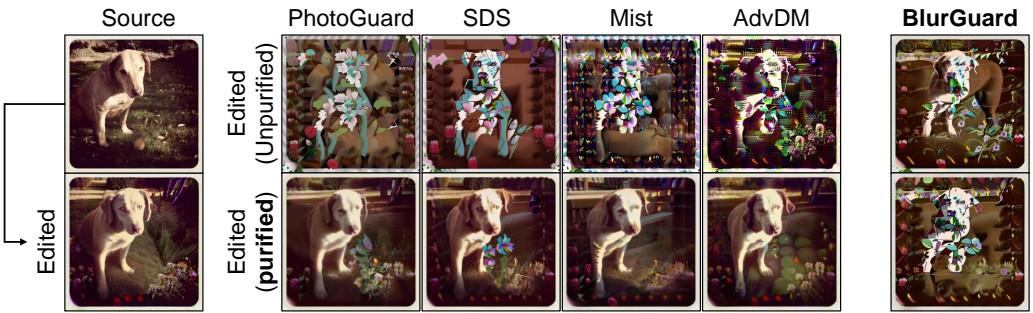

Edit Prompt: "Chesapeake Bay Retriever surrounded by blooming flowers in a garden"

(a)

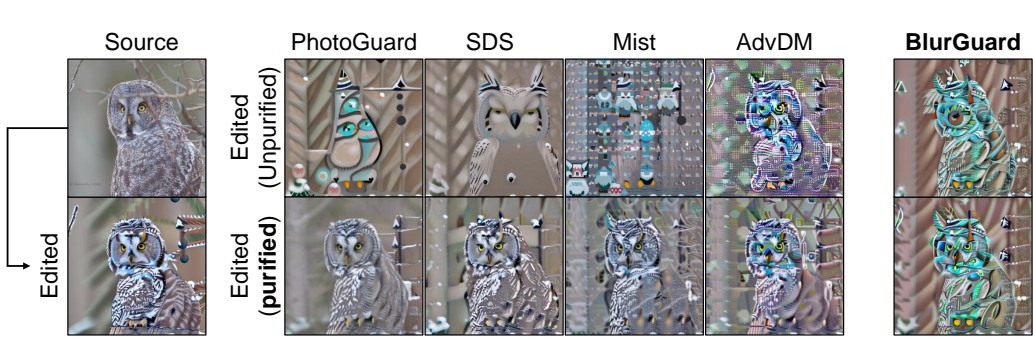

Edit Prompt: "An owl on a snowy day"

(b)

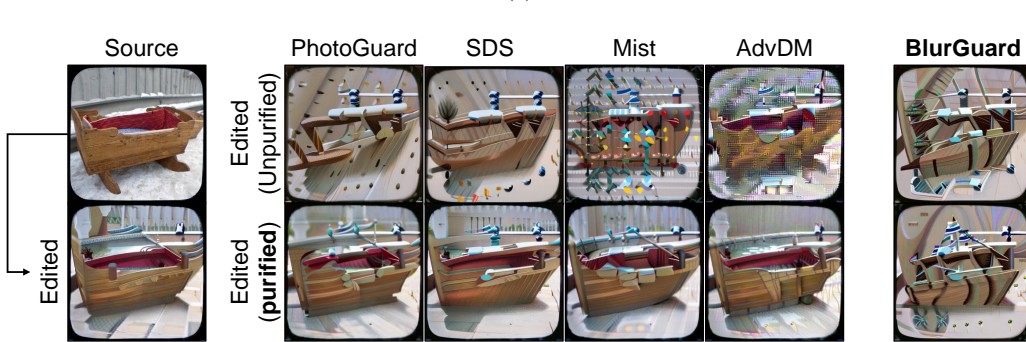

Edit Prompt: "A wooden ship by the pool"

(c)

Figure 10: Qualitative comparison on black-box transfer, where each protection is crafted using SD-v1.4 and tested on SD-v2.1.

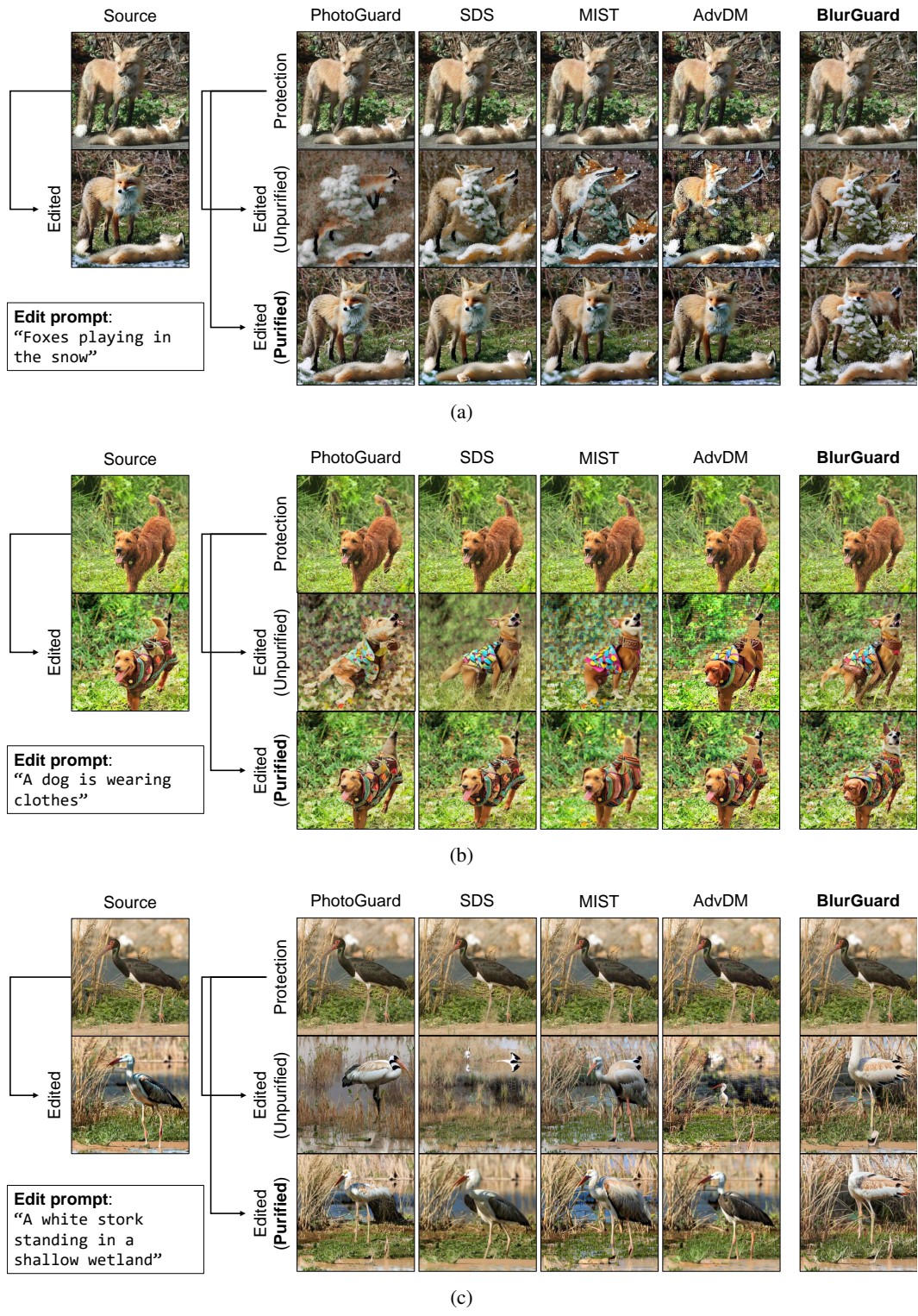

Figure 11: Qualitative results on image-to-image generation under $\ell_\infty$ protection within $\epsilon = \frac{8}{255}$. While all baselines fail to protect the image after purification, only BlurGuard remains robust after purification, generating unrealistic images.

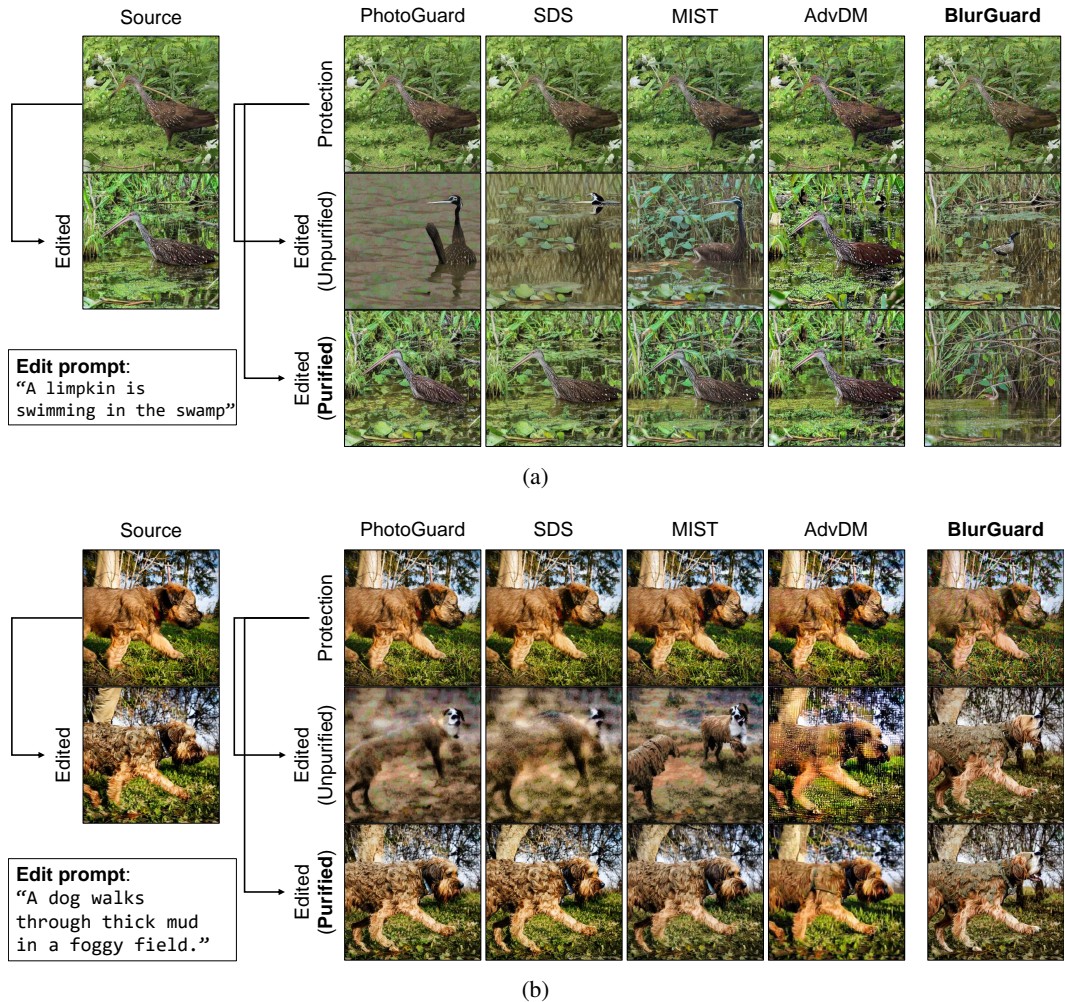

Figure 12: Qualitative results on image-to-image generation under $\ell_\infty$ protection within $\epsilon = \frac{16}{255}$. While all baselines fail to protect the image after purification, only BlurGuard remains robust after purification, generating unrealistic images.

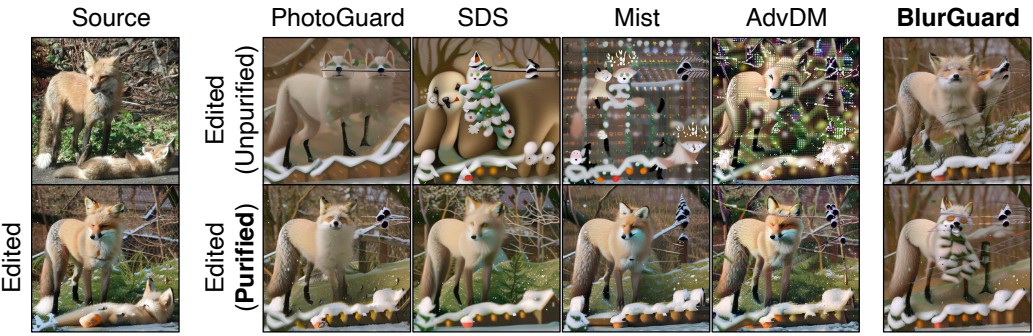

Figure 13: Qualitative comparison of black-box transfer under $\ell_\infty$ protection within $\epsilon = \frac{16}{255}$, where each protection is crafted using SD-v1.4 while tested to SD-v2.1.

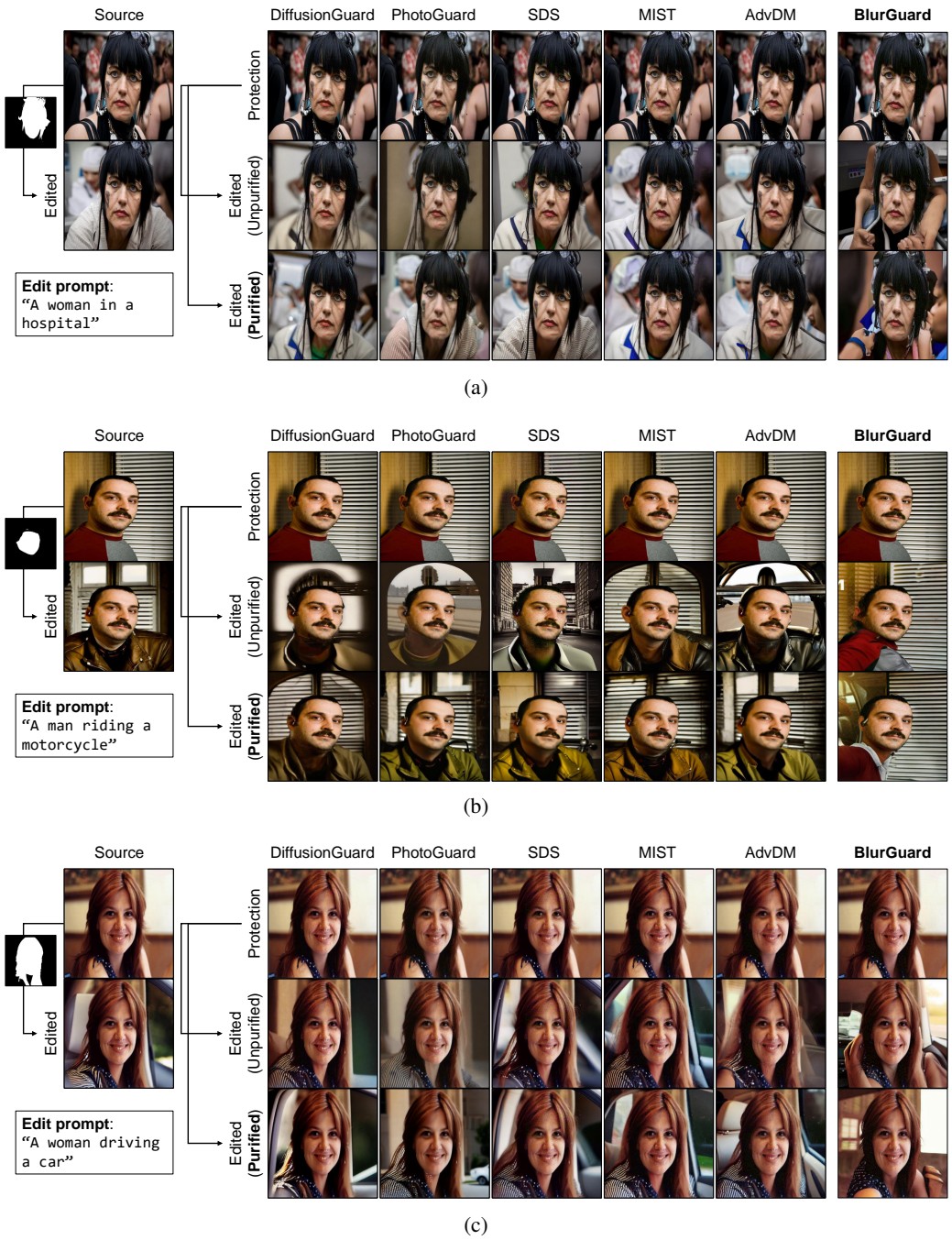

Figure 14: Qualitative inpainting results within $\epsilon = \frac{16}{255}$ . Baselines are eventually purified and produce realistic edits, whereas BlurGuard disrupts inpainting.

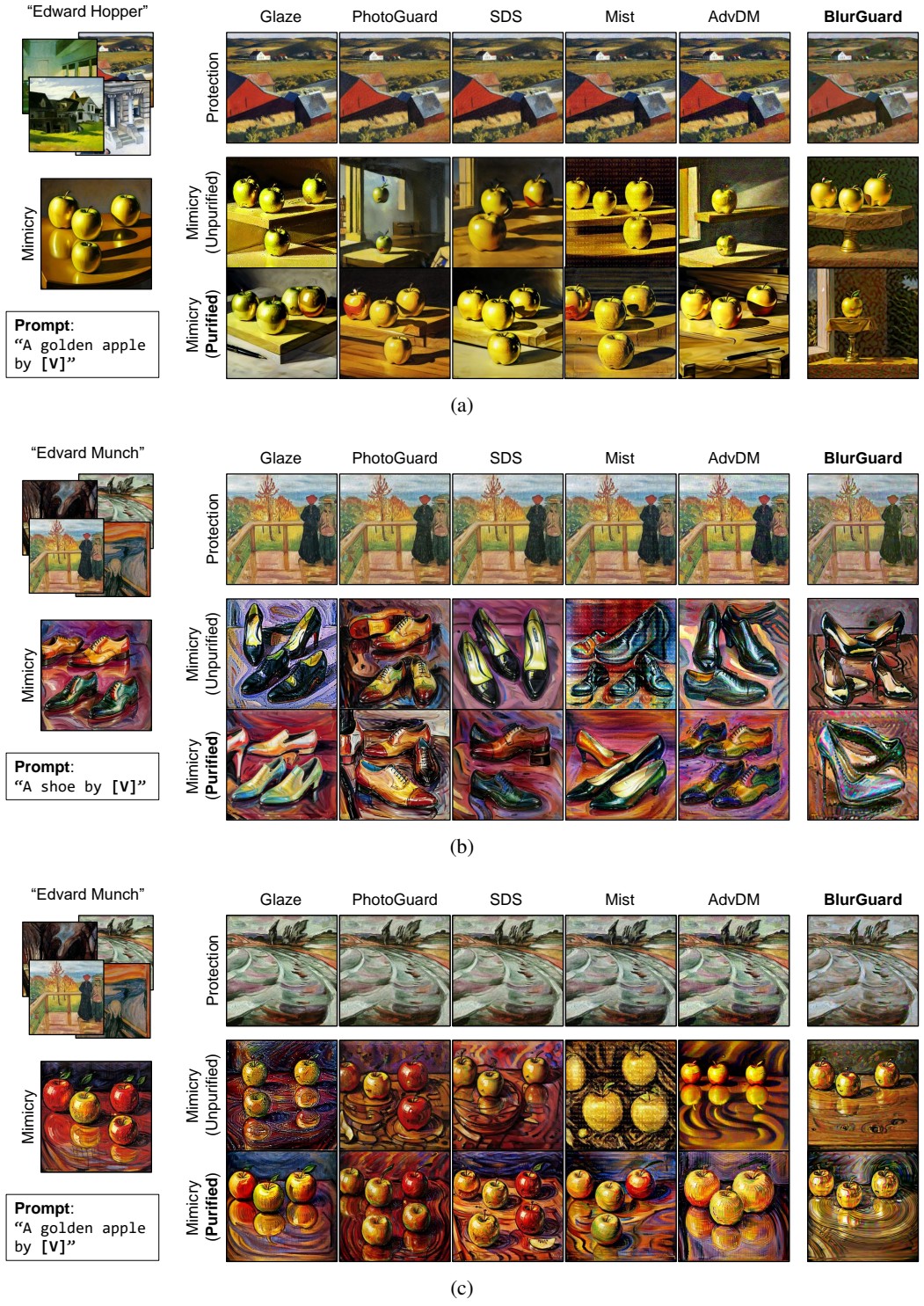

Figure 15: Qualitative results on textual inversion within $\epsilon = \frac{16}{255}$. BlurGuard introduces artifacts that block style extraction; baselines allow near-identical styles.

# F    Individual Effects of Tested Purifications

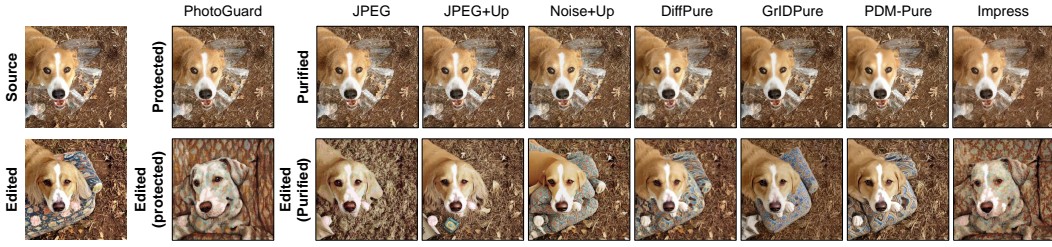

Edit prompt: "A dog sitting on a blue couch"

Figure 16: Qualitative examples of various purified images and the corresponding generated images after applying PhotoGuard protection. The notation "Up" indicates upscaling.

Following our definition of worst-case effectiveness, we test seven different purification methods (*viz.*, as shown in Figure 16) per sample and select the result that deviated the least from the original edited image. Here, we present the detailed results of each noise purification method. Specifically, we evaluate across all eight cases: when each of the seven purification methods is applied individually and when no purification is applied. Table 19 presents the detailed results for the image-to-image generation task, Table 20 corresponds to the inpainting task, and Table 21 reports the results for the textual inversion task.

Although all baseline protections are readily bypassed by the purification tools we evaluate, BlurGuard remains robust against each one of them. Furthermore, we observed that the protection effectiveness of BlurGuard occasionally increases after purification, as reported in Table 19, Table20, and Table 21. This counterintuitive observation emphasizes the resilience of BlurGuard to existing purification techniques. Since BlurGuard relies minimally on high-frequency components as shown in Figure 2a and Figure 8, purification methods that aggressively suppress high frequencies can cause the purified image to deviate further from the source image as illustrated in Figure 2b.

| | **Before Purification** | | | | | **Noise + Upscaling** | | | | |
| Method | **FID ↑** | **LPIPS ↑** | **SSIM ↓** | **PSNR ↓** | **IA ↓** | **FID ↑** | **LPIPS ↑** | **SSIM ↓** | **PSNR ↓** | **IA ↓** |
|---|---|---|---|---|---|---|---|---|---|---|
| PhotoGuard [77] | 201.96 | 0.68 ± 0.10 | 0.38 ± 0.09 | 28.1 ± 0.26 | 0.79 ± 0.08 | 103.65 | 0.35 ± 0.11 | 0.65 ± 0.10 | 29.2 ± 0.70 | 0.90 ± 0.07 |
| AdvDM [49] | 255.55 | 0.69 ± 0.12 | 0.39 ± 0.05 | 28.6 ± 0.41 | 0.78 ± 0.08 | 109.07 | 0.40 ± 0.10 | 0.58 ± 0.08 | 29.0 ± 0.49 | 0.89 ± 0.06 |
| Mist [48] | 233.71 | 0.60 ± 0.08 | 0.42 ± 0.07 | 28.1 ± 0.25 | 0.75 ± 0.09 | 101.25 | 0.37 ± 0.10 | 0.62 ± 0.09 | 29.0 ± 0.62 | 0.88 ± 0.07 |
| SDS [96] | 189.00 | 0.67 ± 0.12 | 0.40 ± 0.10 | 28.2 ± 0.31 | 0.79 ± 0.08 | 97.42 | 0.33 ± 0.09 | 0.66 ± 0.10 | 29.4 ± 0.87 | 0.90 ± 0.06 |
| **BlurGuard (Ours)** | 116.21 | 0.36 ± 0.11 | 0.65 ± 0.11 | 28.5 ± 0.30 | 0.89 ± 0.06 | 119.68 | 0.40 ± 0.10 | 0.61 ± 0.10 | 28.4 ± 0.25 | 0.88 ± 0.06 |
| | **JPEG** | | | | | **JPEG + Upscaling** | | | | |
| Method | **FID ↑** | **LPIPS ↑** | **SSIM ↓** | **PSNR ↓** | **IA ↓** | **FID ↑** | **LPIPS ↑** | **SSIM ↓** | **PSNR ↓** | **IA ↓** |
| PhotoGuard [77] | 141.38 | 0.51 ± 0.11 | 0.50 ± 0.07 | 28.7 ± 0.37 | 0.85 ± 0.08 | 105.08 | 0.36 ± 0.12 | 0.64 ± 0.09 | 29.1 ± 0.51 | 0.88 ± 0.08 |
| AdvDM [49] | 217.45 | 0.59 ± 0.13 | 0.45 ± 0.06 | 28.7 ± 0.39 | 0.83 ± 0.07 | 130.22 | 0.36 ± 0.12 | 0.64 ± 0.09 | 29.1 ± 0.51 | 0.88 ± 0.08 |
| Mist [48] | 161.10 | 0.50 ± 0.10 | 0.50 ± 0.07 | 28.7 ± 0.36 | 0.83 ± 0.08 | 124.02 | 0.42 ± 0.11 | 0.59 ± 0.09 | 28.9 ± 0.42 | 0.86 ± 0.08 |
| SDS [96] | 134.10 | 0.47 ± 0.11 | 0.54 ± 0.09 | 29.1 ± 0.52 | 0.85 ± 0.08 | 91.48 | 0.31 ± 0.09 | 0.68 ± 0.10 | 29.7 ± 0.80 | 0.91 ± 0.06 |
| **BlurGuard (Ours)** | 107.88 | 0.36 ± 0.10 | 0.67 ± 0.11 | 28.5 ± 0.29 | 0.90 ± 0.05 | 116.09 | 0.37 ± 0.10 | 0.66 ± 0.10 | 28.4 ± 0.26 | 0.89 ± 0.06 |
| | **DiffPure** | | | | | **GrIDPure** | | | | |
| Method | **FID ↑** | **LPIPS ↑** | **SSIM ↓** | **PSNR ↓** | **IA ↓** | **FID ↑** | **LPIPS ↑** | **SSIM ↓** | **PSNR ↓** | **IA ↓** |
| PhotoGuard [77] | 92.21 | 0.35 ± 0.09 | 0.69 ± 0.12 | 29.7 ± 1.10 | 0.92 ± 0.05 | 128.02 | 0.34 ± 0.10 | 0.68 ± 0.12 | 28.8 ± 0.69 | 0.88 ± 0.07 |
| AdvDM [49] | 98.27 | 0.37 ± 0.11 | 0.69 ± 0.11 | 29.9 ± 1.02 | 0.91 ± 0.06 | 123.62 | 0.40 ± 0.11 | 0.68 ± 0.08 | 28.8 ± 0.74 | 0.89 ± 0.06 |
| Mist [48] | 90.96 | 0.37 ± 0.10 | 0.69 ± 0.11 | 29.6 ± 0.85 | 0.91 ± 0.06 | 140.20 | 0.39 ± 0.10 | 0.64 ± 0.10 | 28.6 ± 0.71 | 0.86 ± 0.07 |
| SDS [96] | 96.85 | 0.34 ± 0.09 | 0.70 ± 0.11 | 30.1 ± 1.24 | 0.92 ± 0.05 | 114.69 | 0.32 ± 0.09 | 0.69 ± 0.11 | 28.7 ± 0.77 | 0.89 ± 0.07 |
| **BlurGuard (Ours)** | 118.77 | 0.46 ± 0.10 | 0.65 ± 0.11 | 28.5 ± 0.30 | 0.88 ± 0.06 | 133.24 | 0.41 ± 0.10 | 0.66 ± 0.10 | 28.5 ± 0.50 | 0.87 ± 0.06 |
| | **Impress** | | | | | **PDM-Pure** | | | | |
| Method | **FID ↑** | **LPIPS ↑** | **SSIM ↓** | **PSNR ↓** | **IA ↓** | **FID ↑** | **LPIPS ↑** | **SSIM ↓** | **PSNR ↓** | **IA ↓** |
| PhotoGuard [77] | 173.01 | 0.62 ± 0.10 | 0.41 ± 0.08 | 28.2 ± 0.29 | 0.81 ± 0.08 | 124.20 | 0.36 ± 0.10 | 0.69 ± 0.12 | 29.6 ± 1.10 | 0.89 ± 0.06 |
| AdvDM [49] | 197.88 | 0.66 ± 0.10 | 0.19 ± 0.06 | 28.4 ± 0.47 | 0.81 ± 0.08 | 109.36 | 0.36 ± 0.11 | 0.70 ± 0.11 | 30.0 ± 1.16 | 0.89 ± 0.06 |
| Mist [48] | 209.25 | 0.63 ± 0.08 | 0.22 ± 0.09 | 28.1 ± 0.21 | 0.77 ± 0.09 | 121.58 | 0.38 ± 0.11 | 0.68 ± 0.12 | 29.4 ± 0.81 | 0.88 ± 0.06 |
| SDS [96] | 156.37 | 0.59 ± 0.11 | 0.44 ± 0.08 | 28.5 ± 0.35 | 0.83 ± 0.07 | 107.19 | 0.35 ± 0.09 | 0.69 ± 0.11 | 30.1 ± 1.20 | 0.90 ± 0.05 |
| **BlurGuard (Ours)** | 114.12 | 0.40 ± 0.11 | 0.60 ± 0.10 | 28.5 ± 0.27 | 0.88 ± 0.07 | 149.44 | 0.47 ± 0.11 | 0.64 ± 0.12 | 28.4 ± 0.29 | 0.85 ± 0.07 |

Table 19: Detailed results on image-to-image generation under $\ell_\infty$ protection within $\epsilon = \frac{16}{255}$. *Before purification* represents the protection of the editing result without any purification, while the others show the protection after applying seven different noise purification techniques before editing.

**Table 20 — Before Purification / Noise + Upscaling**

| Method | FID ↑ | LPIPS ↑ | SSIM ↓ | PSNR ↓ | IA ↓ | FID ↑ | LPIPS ↑ | SSIM ↓ | PSNR ↓ | IA ↓ |
|---|---|---|---|---|---|---|---|---|---|---|
| PhotoGuard [77] | 171.02 | 0.61 ± 0.08 | 0.46 ± 0.12 | 28.4 ± 0.22 | 0.82 ± 0.08 | 142.39 | 0.48 ± 0.11 | 0.57 ± 0.12 | 28.8 ± 0.47 | 0.87 ± 0.08 |
| AdvDM [49] | 140.15 | 0.42 ± 0.12 | 0.61 ± 0.14 | 29.6 ± 0.58 | 0.88 ± 0.08 | 127.54 | 0.39 ± 0.11 | 0.63 ± 0.11 | 29.2 ± 0.61 | 0.89 ± 0.08 |
| Mist [48] | 131.42 | 0.44 ± 0.10 | 0.59 ± 0.10 | 29.2 ± 0.42 | 0.88 ± 0.06 | 124.66 | 0.39 ± 0.11 | 0.64 ± 0.12 | 29.2 ± 0.57 | 0.89 ± 0.07 |
| SDS [96] | 148.46 | 0.46 ± 0.13 | 0.60 ± 0.14 | 29.2 ± 0.69 | 0.86 ± 0.07 | 137.87 | 0.50 ± 0.10 | 0.56 ± 0.11 | 28.8 ± 0.42 | 0.86 ± 0.07 |
| DiffusionGuard [18] | 164.81 | 0.58 ± 0.09 | 0.50 ± 0.11 | 28.8 ± 0.33 | 0.83 ± 0.08 | 132.70 | 0.41 ± 0.11 | 0.62 ± 0.10 | 29.1 ± 0.51 | 0.88 ± 0.08 |
| **BlurGuard (Ours)** | 162.92 | 0.50 ± 0.10 | 0.59 ± 0.12 | 28.8 ± 0.59 | 0.80 ± 0.08 | 173.90 | 0.57 ± 0.10 | 0.52 ± 0.12 | 28.4 ± 0.25 | 0.79 ± 0.08 |

**JPEG / JPEG + Upscaling**

| Method | FID ↑ | LPIPS ↑ | SSIM ↓ | PSNR ↓ | IA ↓ | FID ↑ | LPIPS ↑ | SSIM ↓ | PSNR ↓ | IA ↓ |
|---|---|---|---|---|---|---|---|---|---|---|
| PhotoGuard [77] | 142.39 | 0.52 ± 0.10 | 0.53 ± 0.11 | 28.9 ± 0.32 | 0.85 ± 0.08 | 131.93 | 0.40 ± 0.11 | 0.62 ± 0.12 | 29.2 ± 0.48 | 0.87 ± 0.08 |
| AdvDM [49] | 119.48 | 0.37 ± 0.14 | 0.64 ± 0.16 | 29.7 ± 0.63 | 0.89 ± 0.08 | 105.26 | 0.29 ± 0.11 | 0.72 ± 0.13 | 30.0 ± 0.80 | 0.94 ± 0.05 |
| Mist [48] | 118.65 | 0.38 ± 0.11 | 0.64 ± 0.13 | 29.5 ± 0.53 | 0.90 ± 0.07 | 109.37 | 0.31 ± 0.09 | 0.71 ± 0.08 | 29.8 ± 0.57 | 0.92 ± 0.04 |
| SDS [96] | 146.17 | 0.47 ± 0.14 | 0.57 ± 0.18 | 29.2 ± 0.64 | 0.85 ± 0.08 | 148.93 | 0.47 ± 0.14 | 0.54 ± 0.18 | 29.1 ± 0.55 | 0.85 ± 0.08 |
| DiffusionGuard [18] | 153.17 | 0.52 ± 0.12 | 0.52 ± 0.17 | 29.1 ± 0.45 | 0.85 ± 0.09 | 123.32 | 0.38 ± 0.12 | 0.65 ± 0.14 | 29.5 ± 0.57 | 0.89 ± 0.08 |
| **BlurGuard (Ours)** | 172.08 | 0.52 ± 0.11 | 0.56 ± 0.15 | 28.3 ± 0.28 | 0.79 ± 0.08 | 166.08 | 0.52 ± 0.11 | 0.56 ± 0.14 | 28.3 ± 0.25 | 0.80 ± 0.07 |

**DiffPure / GrIDPure**

| Method | FID ↑ | LPIPS ↑ | SSIM ↓ | PSNR ↓ | IA ↓ | FID ↑ | LPIPS ↑ | SSIM ↓ | PSNR ↓ | IA ↓ |
|---|---|---|---|---|---|---|---|---|---|---|
| PhotoGuard [77] | 135.91 | 0.44 ± 0.12 | 0.57 ± 0.16 | 29.0 ± 0.48 | 0.88 ± 0.07 | 146.84 | 0.44 ± 0.12 | 0.60 ± 0.15 | 28.5 ± 0.32 | 0.86 ± 0.08 |
| AdvDM [49] | 134.25 | 0.42 ± 0.11 | 0.60 ± 0.15 | 29.2 ± 0.47 | 0.89 ± 0.08 | 144.25 | 0.42 ± 0.12 | 0.62 ± 0.15 | 28.5 ± 0.31 | 0.87 ± 0.08 |
| Mist [48] | 129.47 | 0.41 ± 0.10 | 0.60 ± 0.13 | 29.1 ± 0.43 | 0.90 ± 0.06 | 142.26 | 0.40 ± 0.10 | 0.63 ± 0.12 | 28.5 ± 0.31 | 0.88 ± 0.07 |
| SDS [96] | 133.78 | 0.43 ± 0.11 | 0.59 ± 0.14 | 29.1 ± 0.41 | 0.88 ± 0.07 | 159.96 | 0.46 ± 0.13 | 0.57 ± 0.17 | 28.4 ± 0.30 | 0.85 ± 0.08 |
| DiffusionGuard [18] | 131.03 | 0.43 ± 0.11 | 0.59 ± 0.15 | 29.1 ± 0.42 | 0.88 ± 0.08 | 147.91 | 0.42 ± 0.12 | 0.61 ± 0.15 | 28.5 ± 0.31 | 0.87 ± 0.07 |
| **BlurGuard (Ours)** | 166.29 | 0.55 ± 0.11 | 0.52 ± 0.16 | 28.5 ± 0.28 | 0.80 ± 0.08 | 173.50 | 0.53 ± 0.12 | 0.56 ± 0.17 | 28.6 ± 0.35 | 0.81 ± 0.08 |

**Impress / PDM-Pure**

| Method | FID ↑ | LPIPS ↑ | SSIM ↓ | PSNR ↓ | IA ↓ | FID ↑ | LPIPS ↑ | SSIM ↓ | PSNR ↓ | IA ↓ |
|---|---|---|---|---|---|---|---|---|---|---|
| PhotoGuard [77] | 154.18 | 0.58 ± 0.08 | 0.47 ± 0.10 | 28.6 ± 0.23 | 0.84 ± 0.07 | 153.37 | 0.46 ± 0.10 | 0.61 ± 0.12 | 29.2 ± 0.46 | 0.85 ± 0.08 |
| AdvDM [49] | 140.84 | 0.44 ± 0.11 | 0.59 ± 0.12 | 29.2 ± 0.40 | 0.88 ± 0.07 | 149.97 | 0.45 ± 0.10 | 0.60 ± 0.13 | 29.3 ± 0.50 | 0.86 ± 0.08 |
| Mist [48] | 131.40 | 0.44 ± 0.10 | 0.59 ± 0.09 | 29.1 ± 0.35 | 0.88 ± 0.06 | 145.65 | 0.44 ± 0.11 | 0.61 ± 0.13 | 29.3 ± 0.52 | 0.86 ± 0.08 |
| SDS [96] | 154.33 | 0.48 ± 0.12 | 0.57 ± 0.14 | 29.0 ± 0.50 | 0.85 ± 0.07 | 156.99 | 0.45 ± 0.11 | 0.60 ± 0.13 | 29.3 ± 0.51 | 0.85 ± 0.07 |
| DiffusionGuard [18] | 161.81 | 0.57 ± 0.09 | 0.51 ± 0.10 | 28.8 ± 0.30 | 0.84 ± 0.07 | 149.49 | 0.46 ± 0.12 | 0.59 ± 0.15 | 29.3 ± 0.56 | 0.86 ± 0.08 |
| **BlurGuard (Ours)** | 168.43 | 0.51 ± 0.09 | 0.56 ± 0.10 | 28.4 ± 0.25 | 0.80 ± 0.09 | 180.71 | 0.57 ± 0.12 | 0.53 ± 0.16 | 28.4 ± 0.26 | 0.79 ± 0.09 |

Table 20: Detailed results on inpainting under $\ell_\infty$ protection within $\epsilon = \frac{16}{255}$. *Before purification* represents the protection of the editing result without any purification, while the others show the protection after applying seven different noise purification techniques before editing.

---

**Table 21 — Before Purification / Noise + Upscaling**

| Method | FID ↑ | LPIPS ↑ | SSIM ↓ | PSNR ↓ | IA ↓ | FID ↑ | LPIPS ↑ | SSIM ↓ | PSNR ↓ | IA ↓ |
|---|---|---|---|---|---|---|---|---|---|---|
| PhotoGuard [77] | 195.57 | 0.72 ± 0.12 | 0.20 ± 0.09 | 28.1 ± 0.30 | 0.81 ± 0.08 | 179.01 | 0.62 ± 0.08 | 0.22 ± 0.08 | 28.2 ± 0.27 | 0.85 ± 0.07 |
| AdvDM [49] | 203.50 | 0.68 ± 0.09 | 0.16 ± 0.07 | 28.2 ± 0.39 | 0.81 ± 0.08 | 176.74 | 0.65 ± 0.09 | 0.17 ± 0.07 | 28.2 ± 0.26 | 0.85 ± 0.06 |
| Mist [48] | 233.24 | 0.70 ± 0.08 | 0.12 ± 0.06 | 28.2 ± 0.31 | 0.78 ± 0.08 | 194.00 | 0.64 ± 0.09 | 0.18 ± 0.07 | 28.2 ± 0.27 | 0.84 ± 0.07 |
| SDS [96] | 203.32 | 0.69 ± 0.09 | 0.20 ± 0.08 | 28.1 ± 0.32 | 0.82 ± 0.07 | 176.89 | 0.61 ± 0.08 | 0.22 ± 0.08 | 28.2 ± 0.29 | 0.87 ± 0.06 |
| **BlurGuard (Ours)** | 190.21 | 0.64 ± 0.10 | 0.22 ± 0.10 | 28.2 ± 0.27 | 0.82 ± 0.08 | 181.89 | 0.63 ± 0.09 | 0.22 ± 0.10 | 28.1 ± 0.22 | 0.83 ± 0.08 |

**JPEG / JPEG + Upscaling**

| Method | FID ↑ | LPIPS ↑ | SSIM ↓ | PSNR ↓ | IA ↓ | FID ↑ | LPIPS ↑ | SSIM ↓ | PSNR ↓ | IA ↓ |
|---|---|---|---|---|---|---|---|---|---|---|
| PhotoGuard [77] | 180.28 | 0.64 ± 0.11 | 0.18 ± 0.07 | 28.2 ± 0.35 | 0.86 ± 0.06 | 177.18 | 0.61 ± 0.09 | 0.23 ± 0.09 | 28.2 ± 0.34 | 0.86 ± 0.07 |
| AdvDM [49] | 188.54 | 0.67 ± 0.09 | 0.17 ± 0.06 | 28.2 ± 0.36 | 0.82 ± 0.08 | 188.88 | 0.66 ± 0.10 | 0.18 ± 0.07 | 28.2 ± 0.32 | 0.84 ± 0.07 |
| Mist [48] | 196.23 | 0.66 ± 0.07 | 0.14 ± 0.07 | 28.2 ± 0.33 | 0.82 ± 0.08 | 204.39 | 0.65 ± 0.09 | 0.15 ± 0.06 | 28.2 ± 0.32 | 0.81 ± 0.08 |
| SDS [96] | 182.47 | 0.63 ± 0.09 | 0.20 ± 0.08 | 28.2 ± 0.34 | 0.85 ± 0.06 | 181.00 | 0.61 ± 0.09 | 0.23 ± 0.10 | 28.2 ± 0.30 | 0.86 ± 0.06 |
| **BlurGuard (Ours)** | 199.35 | 0.63 ± 0.09 | 0.22 ± 0.09 | 28.1 ± 0.38 | 0.82 ± 0.07 | 208.87 | 0.63 ± 0.08 | 0.23 ± 0.10 | 28.1 ± 0.25 | 0.83 ± 0.07 |

**DiffPure / GrIDPure**

| Method | FID ↑ | LPIPS ↑ | SSIM ↓ | PSNR ↓ | IA ↓ | FID ↑ | LPIPS ↑ | SSIM ↓ | PSNR ↓ | IA ↓ |
|---|---|---|---|---|---|---|---|---|---|---|
| PhotoGuard [77] | 200.44 | 0.67 ± 0.08 | 0.22 ± 0.10 | 28.2 ± 0.33 | 0.84 ± 0.06 | 196.18 | 0.63 ± 0.10 | 0.23 ± 0.10 | 28.2 ± 0.37 | 0.84 ± 0.06 |
| AdvDM [49] | 194.36 | 0.70 ± 0.09 | 0.20 ± 0.09 | 28.1 ± 0.25 | 0.82 ± 0.08 | 204.69 | 0.64 ± 0.07 | 0.17 ± 0.07 | 28.2 ± 0.34 | 0.83 ± 0.07 |
| Mist [48] | 201.93 | 0.68 ± 0.07 | 0.20 ± 0.08 | 28.1 ± 0.23 | 0.86 ± 0.05 | 204.09 | 0.66 ± 0.08 | 0.17 ± 0.09 | 28.2 ± 0.28 | 0.82 ± 0.07 |
| SDS [96] | 190.13 | 0.66 ± 0.07 | 0.22 ± 0.10 | 28.2 ± 0.28 | 0.84 ± 0.06 | 191.57 | 0.63 ± 0.09 | 0.22 ± 0.09 | 28.2 ± 0.34 | 0.83 ± 0.09 |
| **BlurGuard (Ours)** | 202.08 | 0.71 ± 0.07 | 0.21 ± 0.09 | 28.0 ± 0.15 | 0.81 ± 0.08 | 204.27 | 0.65 ± 0.10 | 0.22 ± 0.09 | 28.1 ± 0.25 | 0.81 ± 0.07 |

**Impress / PDM-Pure**

| Method | FID ↑ | LPIPS ↑ | SSIM ↓ | PSNR ↓ | IA ↓ | FID ↑ | LPIPS ↑ | SSIM ↓ | PSNR ↓ | IA ↓ |
|---|---|---|---|---|---|---|---|---|---|---|
| PhotoGuard [77] | 188.37 | 0.68 ± 0.09 | 0.19 ± 0.07 | 28.1 ± 0.19 | 0.84 ± 0.07 | 200.25 | 0.65 ± 0.08 | 0.23 ± 0.10 | 28.2 ± 0.40 | 0.84 ± 0.06 |
| AdvDM [49] | 200.67 | 0.67 ± 0.09 | 0.17 ± 0.06 | 28.2 ± 0.29 | 0.81 ± 0.08 | 198.20 | 0.65 ± 0.07 | 0.20 ± 0.09 | 28.3 ± 0.39 | 0.83 ± 0.07 |
| Mist [48] | 256.66 | 0.69 ± 0.08 | 0.11 ± 0.05 | 28.2 ± 0.36 | 0.78 ± 0.08 | 235.53 | 0.66 ± 0.11 | 0.20 ± 0.08 | 28.2 ± 0.40 | 0.83 ± 0.06 |
| SDS [96] | 181.76 | 0.66 ± 0.07 | 0.20 ± 0.07 | 28.1 ± 0.24 | 0.84 ± 0.07 | 189.37 | 0.65 ± 0.09 | 0.23 ± 0.10 | 28.2 ± 0.36 | 0.84 ± 0.06 |
| **BlurGuard (Ours)** | 202.42 | 0.64 ± 0.10 | 0.20 ± 0.08 | 28.2 ± 0.48 | 0.81 ± 0.08 | 221.67 | 0.69 ± 0.09 | 0.21 ± 0.08 | 28.1 ± 0.25 | 0.81 ± 0.06 |

Table 21: Detailed results on textual inversion under $\ell_\infty$ protection within $\epsilon = \frac{16}{255}$. *Before purification* represents the protection of the editing result without any purification, while the others show the protection after applying seven different noise purification techniques before editing.

