# OpenReview forum: "BlurGuard: A Simple Approach for Robustifying Image Protection Against AI-Powered Editing"
_NeurIPS.cc/2025/Conference — NeurIPS 2025 poster_

### Official Review · Reviewer_Zo4E · 2025-06-24

**Clarity:** 3
**Significance:** 3
**Originality:** 3
**Rating:** 5
**Confidence:** 4

**Summary:**

This paper introduces a simple but effective method to enhance the robustness of image protection methods against noise reversal techniques, employing an adaptive Gaussian blur on the noise to adjust the overall frequency spectrum. Unlike previous efforts primarily focused on imperceptibility, this approach emphasizes the irreversibility of adversarial noise, making it difficult to detect as noise.

**Questions:**

*Please refer to the weaknesses mentioned above.*

1. The ablation study indicates that "Per-region adaptation" might contribute less significantly compared to other modules. It would be beneficial to qualitatively show the different blur intensities across different regions for specific cases. Additionally, since the blur intensity depends on the semantic regions, a statistical correlation between specific semantic categories and their corresponding blur intensities may reveal interesting insights into how this module functions.
2. Most inpainting methods mask edited regions during processing. A concern arises if the masked area is a region with strong adversarial noise; it's unclear if the proposed method would remain effective. The inpainting experiments (line 278) in this paper are limited to facial images and focus only on modifying areas around the face. However, real-world inpainting scenarios are far more diverse, with unpredictable manipulation regions. This limited experiment might not fully cover all types of inpainting tasks.
3. More comprehensive details are needed for the ImageNet-Edit dataset. This includes clearer information about the prompts used and a statistic of image categories.

**Ethical Concerns:**

["NO or VERY MINOR ethics concerns only"]

**Final Justification:**

Based on the author's detailed rebuttal, which effectively addressed all of my concerns, and after considering the comments and discussions from other reviewers, I have decided to maintain my original score. I recommend the paper for acceptance.

**Limitations:**

Yes

**Paper Formatting Concerns:**

No paper formatting concerns.

**Quality:**

3

**Strengths And Weaknesses:**

**Strengths**:
1. The paper offers a novel perspective by focusing on "irreversibility" rather than solely "imperceptibility" for image protection. And the proposed approach is surprisingly simple but effective.
2. The method's effectiveness is thoroughly demonstrated through comprehensive experiments across various AI-generated scenarios and diverse noise purification techniques.

**Weakness**:
1. The ablation study indicates that "Per-region adaptation" might contribute less significantly compared to other modules. It would be beneficial to qualitatively show the different blur intensities across different regions for specific cases. Additionally, since the blur intensity depends on the semantic regions, a statistical correlation between specific semantic categories and their corresponding blur intensities may reveal interesting insights into how this module functions.
2. Most inpainting methods mask edited regions during processing. A concern arises if the masked area is a region with strong adversarial noise; it's unclear if the proposed method would remain effective. The inpainting experiments (line 278) in this paper are limited to facial images and focus only on modifying areas around the face. However, real-world inpainting scenarios are far more diverse, with unpredictable manipulation regions. This limited experiment might not fully cover all types of inpainting tasks.
3. More comprehensive details are needed for the ImageNet-Edit dataset. This includes clearer information about the prompts used and a statistic of image categories.

---

> ### Author Rebuttal · Authors · 2025-07-31
>
> We sincerely appreciate your thoughtful and constructive feedback. We respond to each of your questions one-by-one in what follows. Should you have any additional comments or concerns, we would be happy to discuss or clarify them.
>
> ---
>
> **Q1. Further analysis on per-region adaptation**
>
> Thank you for your thoughtful suggestion. We agree that a further analysis of the per-region adaptation module would enhance clarity. For your information, our extended ablation study in Appendix G provides a qualitative comparison between BlurGuard and the vanilla PhotoGuard, showing how BlurGuard adaptively allocates noise frequencies and produces perceptually natural perturbations. In the final draft, we will incorporate additional qualitative examples (e.g., more BlurGuard perturbations across different region types) and quantitative results (e.g., correlation between the learned $\sigma$ values and semantic regions, as you suggested) to further improve clarity of the module.
>
> ---
>
> **Q2. Inpainting under mask variability**
>
> As you mentioned, our inpainting experiments primarily focus on “region-targeted” protection, where adversarial perturbations are selectively applied to the region intended for editing. We emphasize that this setup is practically motivated, as it reflects realistic misuse scenarios where a sensitive region, e.g., a person’s face, is kept intact while the surrounding areas are manipulated.
>
> That said, we agree that real-world inpainting scenarios may involve unpredictable and diverse masking patterns. Indeed, this challenge has recently attracted attention in the literature; for example, DiffusionGuard [1], which we also include as a baseline, proposes the InpaintGuardBench dataset to benchmark robustness of image protection methods under mask variability.
>
> To further address your concern, we conducted additional experiments on InpaintGuardBench, evaluating BlurGuard across six different masks per image, including both “seen” (used for protection) and “unseen” (novel at test time) masks. As shown in the table below, BlurGuard maintains strong protection performance across these mask variations, suggesting that BlurGuard not only provides robustness on the region-targeted setup but also generalizes well to more diverse inpainting scenarios. We will incorporate this discussion and the additional results into the final draft.
>
>
>
> | InpaintGuardBench | │ | **Naturalness** |  | │ | **Worst-case**| **effect**| |  |
> |---|-|---|---|-|--|---|---|---|
> | **($\varepsilon=16/255$)** | │ | **LPIPS↓** | **SSIM↑** | │ | **FID↑ (Seen)** | **FID↑ (Unseen)** |  **│ PSNR↓ (Seen)** | **PSNR↓ (Unseen)** |
> | PhotoGuard     | │ | $\underline{\text{0.03±0.02}}$ | 0.96±0.02 | │ | 103.83 | 67.05 |│  37.2±14.2 | 35.5±8.2 |
> | AdvDM          | │ | $\underline{\text{0.03±0.02}}$ | 0.97±0.02 | │ | 125.66 | 71.06 |  │$\underline{\text{35.3±10.3}}$ | $\underline{\text{35.4±7.0}}$ |
> | Mist           | │ | **0.02±0.01** |  $\underline{\text{0.98±0.01}}$ | │ | 112.23 | 58.48 |  │37.1±14.2 | 36.4±9.8 |
> | SDS            | │ | 0.05±0.03 | 0.96±0.02 | │ | 113.30 | $\underline{\text{83.91}}$ |  │37.1±14.2 | 35.6±7.2 |
> | DiffusionGuard | │ |  $\underline{\text{0.03±0.02}}$ | 0.97±0.02 | │ | $\underline{\text{134.25}}$ | 61.32 |  │37.1±14.2 | 35.9±9.9 |
> | **BlurGuard**  | │ | $\underline{\text{0.03±0.02}}$ | **0.99±0.01** | │ | **140.82** | **90.32** |  │**34.9±17.0** | **35.3±8.9** |
>
>
>
>
>
>
>
>
>
>
> ---
>
> **Q3. ImageNet-Edit: Additional details needed**
>
> Thank you for the suggestion. We agree that providing additional clarification on the ImageNet-Edit dataset we constructed would enhance the reproducibility of our work. For your information, we note that we have included the full dataset as part of our supplementary material submission, along with the complete set of prompts used. We also plan to publicly release the dataset upon acceptance to further support reproducibility.
>
> ---
>
> [1] Choi et al., DiffusionGuard: A Robust Defense Against Malicious Diffusion-based Image Editing. ICLR 2025.
>
> ---

---

> > ### Comment · Reviewer_Zo4E · 2025-08-02
> >
> > Thank you for the detailed rebuttal. I have no further questions and will take these updates into account.

---

> > > ### Author Response · Authors · 2025-08-02
> > >
> > > Dear Reviewer Zo4E,
> > >
> > > Thank you for your response and for considering our updates.
> > >
> > > We are glad to hear that our rebuttal addressed your questions.
> > >
> > > Should any additional input be helpful, please let us know.
> > >
> > > Best regards, \
> > > Authors

---

### Official Review · Reviewer_jjT1 · 2025-06-29

**Clarity:** 4
**Significance:** 4
**Originality:** 4
**Rating:** 5
**Confidence:** 3

**Summary:**

The paper tackles the problem that existing adversarial image-protection schemes are easily neutralised by simple post-processing such as JPEG compression or diffusion-based “purification”. The authors observe that many prior perturbations concentrate energy in high-frequency bands and are therefore easy to detect and strip away in the frequency domain. BlurGuard addresses this with two ingredients: (1) Per-region adaptive Gaussian blurring: using SAM masks, different σ values are learned for each semantic region so the perturbation’s spectrum better follows the underlying content. (2) Power-spectrum regularisation: an ℓ∞ penalty on the log-ratio between the RAPSD of the original and protected images keeps the overall spectrum “natural”.

**Questions:**

Please review the Weaknesses section. If all of my concerns are fully resolved, I’ll gladly raise my rating; if the assessment is still lacking, I may lower it instead.

**Ethical Concerns:**

["NO or VERY MINOR ethics concerns only"]

**Final Justification:**

Thank you very much for your effort and thorough response.

I have carefully read all the review comments as well as the authors’ responses. I believe the authors have adequately addressed my concerns, and I have raised my score.

**Limitations:**

yes

**Quality:**

4

**Strengths And Weaknesses:**

## Strengths
1. Re-framing protection as spectral naturalness instead of pure imperceptibility is simple yet insightful.
2. The paper is well written with clear figures.

## Weaknesses
1. The testing primarily relies on outdated UNet-based diffusion models. Experiments on modern models with MMDIT architecture and flow-matching training paradigms, such as SD3.5 [1], FLUX.Fill [2], or Step1X-Edit [3], would strengthen the evaluation.

2. The authors should consider including a section on additional related work, as some studies, such as VINE [4] and Robust-Wide [5], focus on enhancing image watermark robustness to protect images against editing. These works are relevant to the authors’ research, particularly those analyzing image editing from a frequency perspective.

3. The transferability analysis lacks persuasiveness, as SD 1.4 shares the same architecture and some training data with SD 2.1. Demonstrating that image protections optimized for SD-v1.4 are effective against models like FLUX.Fill [2] or Step1X-Edit [3] would provide more robust evidence of transferability.

[1] [SD3.5](https://huggingface.co/stabilityai/stable-diffusion-3.5-large)

[2] [Flux.Fill](https://huggingface.co/black-forest-labs/FLUX.1-Fill-dev)

[3] [Step1X-Edit](https://github.com/stepfun-ai/Step1X-Edit)

[4] [Robust Watermarking Using Generative Priors Against Image Editing: From Benchmarking to Advances](https://arxiv.org/abs/2410.18775)

[5] [Robust-Wide: Robust Watermarking against Instruction-driven Image Editing](https://arxiv.org/abs/2402.12688)

---

> ### Author Rebuttal · Authors · 2025-07-31
>
> We sincerely appreciate your thoughtful and constructive feedback. We respond to each of your questions one-by-one in what follows. Should you have any additional comments or concerns, we would be happy to discuss or clarify them.
>
> ---
>
> **Q1. BlurGuard with MMDiT-based models**
>
> Thank you for the suggestion. We agree that including evaluations on more recent models, e.g., the MMDiT-based architectures with flow-matching you mentioned, would further strengthen our work. Following your suggestion, we have extended our experiments to two additional models, viz., SD-v3.5 [1] and FLUX.1-dev [2], as shown in the following two tables. Overall, the results confirm again that BlurGuard offers the most effective protection against worst-case purification attempts, while also demonstrating the high adaptability of our framework. We will incorporate these additional results into the final draft, and continue to extend our evaluation to other models, e.g., Step1X-Edit [3] you also mentioned.
>
> **Table 1**: Comparison of image protection on ImageNet-Edit, tested on SD-v3.5 [1].
>
> | SD-v3.5 | │ | **Naturalness** |  |  | │ | **Worst** | **case effect** |  |  |  |
> |---|---|---|---|---|---|---|---|---|---|---|
> | **($\varepsilon=16/255$)** | │ | **LPIPS↓** | **SSIM↑** | **PSNR↑** | │ | **FID↑** | **LPIPS↑** | **SSIM↓** | **PSNR↓** | **I.A.↓** |
> | PhotoGuard    | │ | **0.24±0.09**  | 0.79±0.09 | **31.5±1.30** | │ | $\underline{\text{121.58}}$ | 0.45±0.08 | 0.48±0.19 | 28.6±0.64 | **0.88±0.05** |
> | AdvDM         | │ | 0.28±0.10     | 0.77±0.08   | 29.5±0.26 | │ | 116.36      | 0.42±0.08   | 0.48±0.19   | 29.0±0.94   | $\underline{\text{0.89±0.05}}$ |
> | Mist          | │ | $\underline{\text{0.25±0.09}}$   | 0.80±0.09 | $\underline{\text{30.9±1.14}}$   | │ | 118.91 | 0.45±0.08 | 0.48±0.19 | 28.6±0.64 | **0.88±0.05** |
> | SDS           | │ | $\underline{\text{0.25±0.11}}$   | $\underline{0.86±0.08}$ | 29.5±0.32 | │ | 119.20 | $\underline{\text{0.46±0.08}}$ | $\underline{\text{0.47±0.19}}$ | $\underline{\text{28.5±0.56}}$ | **0.88±0.05** |
> | **BlurGuard** | │ |$\underline{\text{0.25±0.10}}$ | **0.88±0.13** | 30.4±1.97 | │ | **125.32** | **0.48±0.07** | **0.46±0.18** | **28.4±0.46** | **0.88±0.05** |
>
>
>
>
> **Table 2**: Comparison of image protection on ImageNet-Edit, tested on FLUX.1-dev [2].
>
> | FLUX.1-dev               | │ | **Naturalness**      |   |   | │ | **Worst**       | **case effect** |   |   |   |
> |-----------------------------|---|----------------------|---|---|---|-----------------|-----------------|---|---|---|
> | **($\varepsilon=16/255$)**  | │ | **LPIPS↓**           | **SSIM↑** | **PSNR↑** | │ | **FID↑**        | **LPIPS↑**       | **SSIM↓** | **PSNR↓** | **I.A.↓** |
> | PhotoGuard                  | │ | **0.21±0.10**        | 0.86±0.07 | 32.1±0.33 | │ | 80.22           | 0.28±0.06        | $\underline{\text{0.68±0.11}}$ | 29.5±0.31  | 0.91±0.05  |
> | AdvDM                       | │ | 0.27±0.10            | 0.85±0.07 | $\underline{\text{32.6±1.10}}$ | │ | 83.42           | 0.28±0.06        | $\underline{\text{0.68±0.12}}$ | 28.8±0.72  | $\underline{\text{0.90±0.05}}$ |
> | Mist                        | │ | 0.26±0.09            | 0.87±0.08 | 32.3±1.40 | │ | $\underline{\text{84.10}}$ | $\underline{\text{0.29±0.07}}$ | **0.67±0.12**        | 29.7±0.60  | $\underline{\text{0.90±0.04}}$ |
> | SDS                         | │ | 0.24±0.10            | $\underline{\text{0.88±0.08}}$ | 32.2±0.80 | │ | 81.65           | 0.28±0.06        | $\underline{\text{0.68±0.11}}$ | $\underline{\text{28.6±0.48}}$ | $\underline{\text{0.90±0.05}}$ |
> | **BlurGuard**               | │ | $\underline{\text{0.23±0.11}}$ | **0.89±0.07** | **33.4±2.90** | │ | **85.21**       | **0.30±0.10**   | **0.67±0.13**        | **28.4±0.33** | **0.89±0.05** |
>
>
> ---
>
> **Q2. Additional related work**
>
> Thank you for suggesting additional related works. For your information, we note that Appendix A.1 already includes a brief discussion on watermarking-based approaches, including the VINE work [4] you referred to. Nevertheless, we agree that providing a dedicated discussion on frequency-based methods explored in other lines of work, e.g., Robust-Wide [5] you mentioned, would help better contextualize our contributions. We will revise our manuscript to include an additional discussion of these related works in the main text of the final draft.
>
> ---
>
> **Q3. Transferability analysis to different architectures, e.g., MMDiT?**
>
> Following your suggestion, we have additionally conducted black-box transfer experiments from SD-v1.4 to SD-v3.5 and FLUX.1-dev; both of which are based on MMDiT with flow-matching training, i.e., significantly diverging from SD-v1.4 in both architecture and training scheme. From the results summarized in the table below, we observe two key points: (a) BlurGuard consistently improves performance over the vanilla PhotoGuard across all metrics, showing the clear effectiveness of BlurGuard to enhance transferability; and (b) BlurGuard remains either the most effective or competitive across the board, further supporting general applicability of our approach. These observations confirm that our proposed framework offers a promising direction for improving the transferability of adversarial image protection methods. We will incorporate these additional results and the respective discussion in the final draft.
>
> ImageNet-Edit|│|**SD‑v1.4**|$\ \ \ \ \ \rightarrow$|**SD‑v3.5**|||│|**SD‑v1.4**|$\ \ \ \ \ \rightarrow$|**FLUX1‑dev**|||
> |---|---|:---|:---|:---|:---|:---|:-:|:---|:---|:---|:---|:---|
> |**($\varepsilon=16/255$)**|│|**FID↑**|**LPIPS↑**|**SSIM↓**|**PSNR↓**|**I.A.↓**|│|**FID↑**|**LPIPS↑**|**SSIM↓**|**PSNR↓**|**I.A.↓**|
> |PhotoGuard|│|112.58|0.40±0.09|0.48±0.18|29.1±0.85|0.90±0.04|│|$\underline{\text{72.60}}$|**0.26±0.08**|0.80±0.07|30.7±0.95|0.95±0.03|
> |AdvDM|│|98.25|0.34±0.10|0.71±0.09|29.9±1.04|0.92±0.04|│|71.74|**0.26±0.08**|**0.78±0.06**|30.6±1.06|$\underline{\text{0.94±0.04}}$|
> |Mist|│|**120.17**|0.43±0.08|0.47±0.19|28.8±0.83|$\underline{\text{0.89±0.05}}$|│|63.59|$\underline{\text{0.21±0.06}}$|0.80±0.06|$\underline{\text{30.3±0.87}}$|0.95±0.03|
> |SDS|│|102.12|0.41±0.09|0.52±0.14|29.4±0.88|$\underline{\text{0.89±0.05}}$|│|58.59|0.19±0.06|0.81±0.07|31.0±1.37|0.96±0.03|
> |**BlurGuard**|│|$\underline{\text{117.71}}$|**0.46±0.08**|**0.46±0.19**|**28.4±0.54**|**0.88±0.05**|│|**74.34**|**0.26±0.09**|$\underline{\text{0.79±0.09}}$|**29.0±0.69**|**0.93±0.04**|
>
> ---
>
> [1] stabilityai/stable-diffusion-3.5-medium \
> [2] black-forest-labs/FLUX.1-dev \
> [3] Liu et al., Step1X-Edit: A Practical Framework for General Image Editing, 2025. \
> [4] Lu et al., Robust Watermarking Using Generative Priors Against Image Editing: From Benchmarking to Advances, ICLR 2025. \
> [5] Hu et al., Robust-Wide: Robust Watermarking against Instruction-driven Image Editing, ECCV 2024.
>
> ---

---

> > ### Comment · Reviewer_jjT1 · 2025-08-01
> >
> > Thank you very much for your effort and thorough response. I can easily imagine that conducting additional experiments must have been quite challenging, and I truly appreciate your dedication.
> >
> > I have carefully read all the review comments as well as the authors’ responses. I believe the authors have adequately addressed my concerns, and I am inclined to raise my score. I would also like to observe the ongoing discussions between the authors and the other reviewers before making a final decision.
> >
> > Thank you again.

---

> ### Author Response · Authors · 2025-08-01
>
> Dear Reviewer jjT1,
>
> Thank you very much for your prompt response and encouraging words!
>
> We truly value your thoughtful consideration, and we would be happy to address any further clarification you may need as the discussion continues.
>
> Best regards, \
> Authors

---

### Official Review · Reviewer_Y1wo · 2025-06-30

**Clarity:** 1
**Significance:** 4
**Originality:** 2
**Rating:** 4
**Confidence:** 5

**Summary:**

This paper asserts that existing data poisons for diffusion models are weak to purification-based defenses because they fail to conform to the data prior expected by the diffusion models. They propose that this nonconformity can be identified as a deviation from real (unpoisoned) images in the frequency spectrum, as measured by RAPSD curves. They propose an adaptive per-region blurring operation applied to adversarial noise in order to encourage the adversarial images to align closer to real images. They present experiments, comparing BlurGuard to multiple current poisons and validating their method across multiple defenses. They measure the naturalness (i.e., imperceptibility) and effectiveness (i.e., worst-case performance decrease after purification) of adversarial images.

**Questions:**

I have given multiple suggestions in Strengths and Weaknesses. In particular, I emphasize additional ablation studies and improved clarity in section 4. I think these improvements are necessary to accept the paper.

**Ethical Concerns:**

["NO or VERY MINOR ethics concerns only"]

**Final Justification:**

The authors have answered my questions and addressed my concerns.

**Limitations:**

Yes

**Paper Formatting Concerns:**

Some near unreadable figure labels

**Quality:**

3

**Strengths And Weaknesses:**

- Significance: Protection for personal data or copyright against copying or theft is an important topic. Moreover, existing protections for images are easily circumvented by purification-based (or other) defenses. Research into effective data poisons is necessary if artists and copyright holders are to continue publishing their work in a public setting.
- Originality: BlurGuard has moderate originality. Most poisons in diffusion literature utilize an adversarial component and a perceptibility constraint (e.g., LPIPS in Glaze/Nightshade or a simple norm-ball on adversarial signal). Prior works on poisoning in other domains have explored blurring and sharpening filters as poisons (e.g., https://arxiv.org/abs/2303.04278, https://link.springer.com/chapter/10.1007/978-3-031-73464-9_5). In the diffusion domain, Gaussian blur has been used to model transitions between probability distributions (i.e., heat dissipation vs diffusion) (e.g., https://arxiv.org/pdf/2206.13397). Papers in that area have also utilized spectral density plots for analysis. Regardless, I have not seen these specific components combined for the purpose of poisoning. I think that more exploration into poisons that focus on the frequency-spectrum analysis is necessary, and so BlurGuard is a step in the right direction. The existing ablations in the paper and Appendix G are insightful and valuable. It would be even better if the paper could conduct further ablations to serve as a foundation for further research. Some that come to mind include testing other adversarial objectives instead of just EncoderAttack, analyzing sharpening in addition to blur, and applying blur to the adversarial image (x + delta) instead of just the adversarial noise.
- Clarity: Clarity is a particular weakness of this paper. My main complaint is the organization of sections 3 and 4. Equation 1 has a notational error, as the argmin of L_enc is equal to the MSE objective, but its argmin is not. Equation 4 is introduced and then never used. I think "Per-Region Adaptation" should be introduced before "Learnable Gaussian Blurring" in section 4.2 since masking and blurring are strongly linked in BlurGuard. Notation for distance constraints seems inconsistent, using d(-,-), ball B, or ||-||. Also it seems epsilon and eta are used interchangeably. Perhaps a simplified (shorter) Algorithm 1 should be shown in the main paper because the multi-stage optimization procedure is unintuitive, especially with using dummy deltas during stage 1. Elsewhere, some figures have near-unreadable labels (e.g., Fig. 2, Fig. 4). Fig. 4 and its discussion are confusing: I don't understand what is meant by the "Worst-Case Gap", does the low value of BlurGuard's "Worst-case Gap" indicate that it performs strongly even with a 1/255 adversarial perturbation?
- Quality: The paper is generally well-written, with straightforward introduction and experiments sections, as well as a nicely organized Appendix. I cite the extensive studies in the appendix as a strength. The authors have also benchmarked performance against a large set of modern adversarial methods and purification methods.

---

> ### Author Rebuttal · Authors · 2025-07-31
>
> We sincerely appreciate your thoughtful and constructive feedback. We respond to each of your questions one-by-one in what follows. Should you have any additional comments or concerns, we would be happy to discuss or clarify them.
>
> ---
>
> **Q1. BlurGuard with other adversarial objectives**
>
>
> As you mentioned, BlurGuard is compatible with a variety of adversarial image protection objectives beyond the encoder attack primarily considered in our experiments. Following your suggestion, we have additionally tested BlurGuard in combination with other adversarial objectives, viz., AdvDM [1] and Mist [2]. Specifically, AdvDM aims to directly maximize the denoising loss of the diffusion process, and Mist considers a combined form of encoder and denoising attacks. As shown in the table below, we observe that BlurGuard preserves its effectiveness in improving both naturalness and worst-case effectiveness of both attacks, confirming the broad applicability of our framework across diverse protection objectives. We will incorporate these additional results in the final draft.
>
> |ImageNet-Edit| │ |**Naturalness**|||│|**Worst**|**case effect**||||
> |---|---|---|---|---|---|---|---|---|---|---|
> | **($\varepsilon=16/255$)** | │|**LPIPS↓**|**SSIM↑**|**PSNR↑**|│|**FID↑**|**LPIPS↑**|**SSIM↓**|**PSNR↓**|**I.A.↓**|
> |AdvDM| │|0.36±0.12|0.74±0.09|28.9±0.28|│|98.27|0.30±0.08|0.73±0.09|30.2±1.17|**0.93±0.04**|
> |**AdvDM + BlurGuard**| │|**0.25±0.07**|**0.81±0.09**|**30.1±2.19**|│|**101.84**|**0.35±0.09**|**0.70±0.09**|**28.2±2.03**|**0.93±0.05**|
> |Mist| │|0.35±0.12|0.72±0.10|28.6±0.25|│|90.96|0.30±0.07|0.71±0.09|29.7±0.86|0.93±0.04|
> |**Mist + BlurGuard**| │|**0.33±0.11**|**0.86±0.08**|**32.3±2.21**|│|**137.91**|**0.42±0.10**|**0.63±0.07**|**28.8±0.47**|**0.89±0.06**|
>
> ---
>
> **Q2. Additional ablation study**
>
> We appreciate your insightful suggestions regarding further ablation studies. In response, we have conducted additional experiments based on the following two ablations of BlurGuard: (a) applying blurring directly to the protected image (i.e., $\mathbf{x} + \boldsymbol{\delta}$) rather than only to the perturbation (i.e., $\boldsymbol{\delta}$), and (b) replacing the blurring operation with sharpening, which effectively acts as a high-pass filter. From the results shown in the table below, we observe the following:
> - (a) Blurring $\mathbf{x} + \boldsymbol{\delta}$ has a moderate effect in enhancing robustness, e.g., over the vanilla PhotoGuard. However, it also results in a notable drop in naturalness metrics compared to BlurGuard, as blurring the entire image erases high-frequency details from the image itself and thus compromises perceptual quality. In contrast, BlurGuard can avoid the issue by applying blur only to $\boldsymbol{\delta}$, preserving both robustness and visual fidelity.
> - (b) The adaptive sharpening scheme, implemented via unsharp masking [3], did not provide effective protection after purification. This is because sharpening rather amplifies high-frequency patterns, which can be easily suppressed by known purification techniques, e.g., JPEG compression.
>
> Overall, these results highlight our proposed adaptive blurring scheme as a more effective strategy, achieving both robustness and perceptual naturalness. We will incorporate these results in our final manuscript.
> |ImageNet-Edit| │ |**Naturalness**|||│|**Worst**|**case effect**||||
> |---|---|---|---|---|---|---|---|---|---|---|
> |**($\varepsilon=16/255$)**| │|**LPIPS↓**|**SSIM↑**|**PSNR↑**|│|**FID↑**|**LPIPS↑**|**SSIM↓**|**PSNR↓**|**I.A.↓**|
> |PhotoGuard|│|0.34±0.11|0.70±0.11|28.2±0.32|│|92.21|0.27±0.07|0.73±0.10|29.9±1.07|0.94±0.04|
> |**BlurGuard**|│|**0.21±0.10**|**0.93±0.09**|31.1±2.29|│|**107.88**|**0.32±0.09**|**0.70±0.09**|**28.8±0.42**|**0.92±0.05**|
> |└─blur($\mathbf{x}+\boldsymbol{\delta}$)|│|0.32±0.11|0.79±0.08|**32.3±1.11**|│|98.45|0.30±0.07|0.71±0.10|30.1±1.21|0.94±0.04|
> |└─sharpen($\boldsymbol{\delta}$)|│|0.35±0.12|0.65±0.10|30.9±1.08|│|82.72|0.28±0.08|0.72±0.10|30.4±1.38|0.94±0.04|
>
>
>
>
> ---
>
> **Q3. Editorial comments**
>
> Thank you very much for your detailed reading. We agree that your editorial suggestions will help strengthen the clarity of our manuscript. We will thoroughly review all of your comments and incorporate them into the final draft, e.g., including the following points:
> Equation (1) will be rewritten to separately define $L\_{\mathrm{enc}}$ apart from the $\mathrm{argmin}$ formulation.
> The distance notations will be clarified, e.g., by standardizing the use of $\epsilon$ in place of $\eta$ for consistency.
> A simplified version of Algorithm 1 will be incorporated into the main text.
> The font sizes of Figure 2 and 4 will be increased to improve readability.
>
> ---
>
> **Q4. Figure 4: “Worst-case gap”?**
>
> We use the term “worst-case gap” in Figure 4 to refer to the drop in protection effectiveness (e.g., in LPIPS) between a protected image before and after the strongest purification attempt among those considered. In other words, it measures how much the protection effect is weakened by the strongest purification. Therefore, the small worst-case gaps of BlurGuard in Figure 4 show that its protections remain robust even under aggressive purification attacks. This trend holds consistently across a wide range of protection budgets; e.g., including those from $\epsilon=\tfrac{1}{255}$, as you mentioned, up to $\epsilon=\tfrac{16}{255}$. We will clarify this point in the final draft.
>
> ---
>
> [1] Liang et al., Adversarial Example Does Good: Preventing Painting Imitation from Diffusion Models via Adversarial Examples. ICML 2023. \
> [2] Liang & Wu, Mist: Towards improved adversarial examples for diffusion models. 2023. \
> [3] Polesel et al., Image enhancement via adaptive unsharp masking. IEEE TIP 2000.
>
> ---

---

> ### Author Response · Authors · 2025-08-04
>
> Dear Reviewer Y1wo,
>
> Thank you again for your time and effort in reviewing our manuscript.
>
> As we are now midway through the discussion period, we would like to kindly check if you have any remaining questions or concerns we could help clarify.
>
> We believe we have made a sincere effort to address your earlier comments, and would be happy to receive any additional feedback you may have.
>
> Your insights are invaluable and would greatly help us further strengthen our manuscript.
>
> Best regards, \
> Authors

---

### Official Review · Reviewer_fDxR · 2025-07-01

**Clarity:** 3
**Significance:** 3
**Originality:** 3
**Rating:** 5
**Confidence:** 4

**Summary:**

BlurGuard is a effective image protection method designed to improve robustness against AI-driven image editing. It addresses the vulnerability that prior adversarial noise-based protections can be easily reversed by simple post-processing. BlurGuard’s approach is to apply an adaptive local Gaussian blur to the embedded perturbation to adjust its frequency spectrum, making the protective noise difficult to extract while mitigating image quality loss. Experiments show that this method significantly improves worst-case protection performance under diverse image editing scenarios and noise-removal attacks, and achieves superior image quality compared to existing methods.

**Questions:**

1. Lack of Comparison with the Most Recent Methods. The comparative methods used in the experimental section are primarily from 2023 and do not include the most up-to-date techniques. For example, more recent approaches such as Distraction is All You Need and EditShield could be incorporated to provide a more comprehensive evaluation.
2. Insufficient Explanation in Certain Experimental Results. Some parts of the experimental analysis lack detailed interpretation. For instance, in Table 9, the BlurGuard method exhibits relatively lower performance in terms of PSNR under the "Naturalness" metric, yet no corresponding explanation is provided in the text.
3. Lack of Discussion on Alternative Tampering Methods. BlurGuard is based on adversarial perturbation. If an attacker employs alternative editing techniques, such as methods beyond diffusion models, the protective effectiveness of BlurGuard may be limited. This potential limitation is practically relevant and deserves further discussion.
4. Omission of Acronym Explanations. Certain acronyms are not clearly defined in the main text. For example, the “I.A.” metric in Table 1 is not explicitly explained in the body of the paper. Although it is addressed in the appendix, its meaning should be clarified in the main text or table caption to avoid confusion, especially for first-time readers.

**Ethical Concerns:**

["NO or VERY MINOR ethics concerns only"]

**Final Justification:**

The authors have addressed most of our concerns in the rebuttal process.

**Limitations:**

Lack of Discussion on Alternative Tampering Methods. BlurGuard is based on adversarial perturbation. If an attacker employs alternative editing techniques, such as methods beyond diffusion models, the protective effectiveness of BlurGuard may be limited. This potential limitation is practically relevant and deserves further discussion.

**Quality:**

3

**Strengths And Weaknesses:**

Strengths：
1. Innovativeness of the Work.This study demonstrates a notable degree of innovation. The proposed BlurGuard method adjusts the frequency bandwidth of adversarial noise via adaptive Gaussian blurring, presenting a simple yet effective strategy to enhance the robustness of adversarial image protection. Compared to existing approaches, this method emphasizes alignment with the natural frequency spectrum of images, thereby improving the perceptual naturalness of the protected outputs.
2. Comprehensiveness of the Experimental Evaluation. The experimental section of the paper is well-developed and comprehensive. It provides strong empirical evidence supporting the effectiveness and robustness of BlurGuard. The experiments are rigorously designed and the results are presented in detail, highlighting the advantages of the method under various attack scenarios.
3. Construction of a Dataset. The study acknowledges the lack of publicly available standardized datasets for malicious image editing and accordingly constructs a tailored evaluation dataset. This dataset encompasses a diverse set of images and editing intentions, facilitating consistent and meaningful comparisons across different protection methods.
4. Clarity and Logical Organization of the Paper. The paper is clearly written and logically organized. The arguments are well-structured and coherent, enabling readers to follow the motivation, methodology, and contributions of the work in a systematic and persuasive manner.
Weakness：
1. Lack of Comparison with the Recent Methods.The comparative methods used in the experimental section are primarily from 2023 and do not include the most up-to-date techniques. For example, more recent approaches such as Distraction is All You Need and EditShield could be incorporated to provide a more comprehensive evaluation.
2. Insufficient Explanation in Certain Experimental Results.Some parts of the experimental analysis lack detailed interpretation. For instance, in Table 9, the BlurGuard method exhibits relatively lower performance in terms of PSNR under the "Naturalness" metric, yet no corresponding explanation is provided in the text.
3. Lack of Discussion on Alternative Tampering Methods.BlurGuard is based on adversarial perturbation. If an attacker employs alternative editing techniques, such as methods beyond diffusion models, the protective effectiveness of BlurGuard may be limited. This potential limitation is practically relevant and deserves further discussion.
4. Omission of Acronym Explanations.Certain acronyms are not clearly defined in the main text. For example, the “I.A.” metric in Table 1 is not explicitly explained in the body of the paper. Although it is addressed in the appendix, its meaning should be clarified in the main text or table caption to avoid confusion, especially for first-time readers.

---

> ### Author Rebuttal · Authors · 2025-07-31
>
> We sincerely appreciate your thoughtful and constructive feedback. We respond to each of your questions one-by-one in what follows. Should you have any additional comments or concerns, we would be happy to discuss or clarify them.
>
> ---
>
> **Q1. Additional baselines**
>
> Thank you for suggesting additional baselines. In our experiments, we selected our baselines that are most commonly considered in recent literature on adversarial image protection. We note that we also made an effort to include more recent techniques where possible, e.g., DiffusionGuard (2025) [1] and High-Frequency Anti-DreamBooth (2024) [2], as well as state-of-the-art purification-based defenses like GrID‑Pure (2024) [3] and PDM‑Pure (2024) [4]. In response to your suggestion, we have additionally compared BlurGuard with EditShield [5] you mentioned, although we were unfortunately unable to include “Distraction is All You Need” [6] due to the lack of publicly available code. As summarized in the table below, the results further confirm the strong robustness of BlurGuard as a natural and effective image protection method. We will incorporate these and more results in the final draft.
>
> | MagicBrush | │ | **Naturalness** |  |  | │ | **Worst** | **case effect** |  |  |  |
> |---|---|---|---|---|---|---|---|---|---|---|
> | **($\varepsilon=16/255$)** | │ | **LPIPS↓** | **SSIM↑** | **PSNR↑** | │ | **FID↑** | **LPIPS↑** | **SSIM↓** | **PSNR↓** | **I.A.↓** |
> | PhotoGuard | │ | **0.19±0.10** | 0.74±0.09 | **31.1±0.22** | │ | 96.23 | 0.21±0.09 | 0.73±0.11 | 30.7±1.50 | 0.96±0.03 |
> | AdvDM      | │ | 0.27±0.08     | $\underline{0.76\pm 0.07}$ | 28.9±0.24     | │ | 123.39 | 0.30±0.09 | 0.67±0.13 | 30.2±1.25 | $\underline{0.93\pm 0.04}$ |
> | Mist       | │ | 0.26±0.09     | 0.74±0.08 | 28.6±0.17     | │ | $\underline{123.54}$ | 0.30±0.09 | 0.66±0.12 | $\underline{29.8\pm 0.94}$ | $\underline{0.93\pm 0.04}$ |
> | SDS        | │ | 0.23±0.07     | 0.74±0.07 | 29.1±0.37     | │ | 115.23 | 0.27±0.09 | 0.68±0.13 | 30.3±1.38 | 0.94±0.04 |
> | $\textcolor{blue}{\text{EditShield}}$ | │ | 0.36±0.13 | 0.75±0.09 | $\underline{31.0\pm 0.23}$ | │ | 120.13 | $\underline{0.33\pm 0.09}$ | $\underline{0.65\pm 0.13}$ | 29.9±1.26 | $\underline{0.93\pm 0.04}$ |
> | **BlurGuard** | │ | $\underline{0.20\pm 0.08}$ | **0.89±0.07** | 30.1±1.76 | │ | **138.22** | **0.36±0.10** | **0.64±0.12** | **28.8±0.55** | **0.90±0.06** |
>
>
>
>
>
>
> ---
>
> **Q2. Table 9: Clarification on PSNR results**
>
> We remark that PSNR primarily captures pixel-wise differences to measure naturalness, and accounts less for perceptual relevance; often penalizing even imperceptible changes. BlurGuard intentionally applies stronger perturbations in perceptually insensitive regions to improve robustness while maintaining visual quality. These imperceptible changes can slightly lower PSNR, despite not affecting human perception. This aspect is better reflected in stronger performance on perceptual metrics like SSIM and LPIPS, and is further supported by qualitative results in Figure 11 in Appendix. We will clarify this point in the final version.
>
> ---
>
> **Q3. Discussion on non-diffusion editing methods**
>
> We acknowledge that the transferability of adversarial image protection methods, e.g., to non-diffusion editing methods, is an important step toward increasing their practical applicability. In particular, the question of whether adversarial perturbations crafted for one type of editor (e.g., a diffusion model) can effectively transfer to other types of (black-box) editing tools, e.g., GAN-based models, remains a technically challenging and, to our knowledge, under-explored problem. In this work, we focus on diffusion-based pipelines, as they currently represent the most widely used class of image editing models in both academic and commercial settings. We think extending BlurGuard to cover non-diffusion editors is a promising future direction, and we will reflect this point in the final draft by revising the discussion of limitations (e.g., in Appendix A.2).
>
> ---
>
> **Q4. “I.A.”: Should be defined in the main text**
>
> Thank you for the incisive comment. We will ensure that the I.A. metric is clearly defined and explained in the main text of the final draft.
>
> ---
> [1] Choi et al., DiffusionGuard: A Robust Defense Against Malicious Diffusion-based Image Editing. ICLR 2025. \
> [2] Onikubo et al., High-Frequency Anti-DreamBooth: Robust Defense against Personalized Image Synthesis. ECCV Workshop 2024. \
> [3] Zhao et al., Can Protective Perturbation Safeguard Personal Data from Being Exploited by Stable Diffusion? CVPR 2024. \
> [4] Xue et al., Pixel is a Barrier: Diffusion Models Are More Adversarially Robust Than We Think. 2024. \
> [5] Chen et al., EditShield: Protecting Unauthorized Image Editing by Instruction-guided Diffusion Models. ECCV 2024. \
> [6] Lo et al., Distraction is all you need: Memory-Efficient Image Immunization against Diffusion-Based Image Editing. CVPR 2024.
>
> ---

---

> ### Author Response · Authors · 2025-08-04
>
> Dear Reviewer fDxR,
>
> Thank you again for your time and effort in reviewing our manuscript. We are also delighted and encouraged by your positive review, and we truly appreciate your support for our manuscript.
>
> As we are now midway through the discussion period, we would like to kindly check if you have any remaining questions or concerns we could help clarify.
>
> We believe we have made a sincere effort to address your earlier comments, and would be happy to receive any additional feedback you may have.
>
> Your insights are invaluable and would greatly help us further strengthen our manuscript.
>
> Best regards, \
> Authors

---

> > ### Comment · Reviewer_fDxR · 2025-08-05
> > **Practical Implications for Real-World Applications:**
> >
> > While BlurGuard shows strong robustness against diffusion-based models, the practical relevance of this work could be limited if the defense cannot be easily adapted to these other techniques. The increasing use of GAN-based editors (such as in commercial applications like deepfake generation) raises important questions regarding the applicability of BlurGuard outside of diffusion models. It would be helpful to address how these alternative methods could potentially circumvent BlurGuard’s protections and what specific challenges this creates.

---

> > > ### Author Response · Authors · 2025-08-05
> > >
> > > Dear Reviewer fDxR,
> > >
> > > Thank you for your follow-up and for raising this point before the end of the discussion period, allowing us to address it more thoroughly.
> > >
> > > **Q3-2. BlurGuard on other editing methods, e.g., GANs**
> > >
> > > In response to your concern, we have additionally conducted a black-box transfer experiment from SD-v1.4 to SimSwap [7], a popular GAN-based method for deepfake generation, using the human portrait subset of InpaintGuardBench [8] as the test set. As shown in the table below, BlurGuard continues to demonstrate strong performance in both naturalness and worst-case effectiveness. We find this to be a particularly interesting observation, as it confirms the strong transferability of our approach not only across architectures, but also across fundamentally different generative paradigms, i.e., from diffusion models to GANs. We will incorporate these additional results and the corresponding discussion into the final draft.
> > >
> > > | SD-v1.4 $\rightarrow$ SimSwap | │ | **Naturalness** |  | │ | **Worst** | **case effect** |
> > > |---|---|---|---|---|---|---|
> > > | **($\varepsilon=16/255$)** | │ | **LPIPS↓** | **SSIM↑** | │ | **FID↑** | **PSNR↓** |
> > > | PhotoGuard    | │ | 0.04±0.02 | 0.96±0.02 | │ | 18.34 | 40.0±2.67 |
> > > | AdvDM         | │ | 0.04±0.03 | 0.97±0.01 | │ | 17.16 | 40.4±2.43 |
> > > | Mist          | │ | **0.02±0.01** | $\underline{\text{0.98±0.01}}$ | │ | 14.10 | 41.0±2.43 |
> > > | SDS           | │ | 0.06±0.03 | 0.96±0.02 | │ | $\underline{\text{25.11}}$ | $\underline{\text{39.3±2.44}}$ |
> > > | DiffusionGuard| │ | 0.04±0.02 | 0.97±0.01 | │ | 17.39 | 40.2±2.33 |
> > > | **BlurGuard** | │ | $\underline{\text{0.03±0.02}}$ | **0.99±0.01** | │ | **31.54** | **38.4±2.37** |
> > >
> > > Furthermore, we clarify that our BlurGuard framework is not limited to diffusion-based models even in white-box setups, i.e., it can be readily applied to protect images directly against other editing models. This is because our approach does not rely on any prior specific to diffusion models; rather, it introduces a simple regularization term that can be added to arbitrary loss objectives, exploiting frequency characteristics of adversarial noise shared across models. In other words, we expect that the protective performance of BlurGuard could be further enhanced if it were directly applied to disrupt GAN outputs (e.g., for deepfakes, where GANs remain widely used), although we consider this to be beyond the current scope.
> > >
> > > Thank you once again for your engagement and valuable feedback!
> > >
> > > Best regards, \
> > > Authors
> > >
> > > ---
> > >
> > > [7] Chen et al., SimSwap: An Efficient Framework For High Fidelity Face Swapping. ACM MM 2020. \
> > > [8] Choi et al., DiffusionGuard: A Robust Defense Against Malicious Diffusion-based Image Editing. ICLR 2025.
> > >
> > > ---

---

### Note · Authors · 2025-08-14

Dear Area Chair and Reviewers,

We would like to express our sincere gratitude for your time and effort in reviewing our manuscript. As a final remark of the discussion phase, we provide a brief summary of how we will address the reviewers’ comments in the final draft.

We are delighted that our work has been received positively, based on our understanding, by all reviewers at the conclusion of the discussion phase. As highlighted in the reviews, our work proposes a simple yet effective framework (fDxR, Zo4E, jjT1) for robust image protection, grounded in a novel perspective (fDxR, Zo4E, jjT1), supported by extensive experiments (fDxR, Y1wo, Zo4E), and presented clearly (fDxR, Y1wo).

We have conducted additional experiments in response to the reviewers’ feedback, and the key updates for the final draft are as follows:

- Additional baseline for the instruction-based editing task, viz., EditShield (fDxR)
- BlurGuard with other adversarial objectives, viz., Mist and AdvDM (Y1wo)
- Evaluation using MMDiT-based flow models, e.g., SD-v3.5 and FLUX.1-dev (jjT1)
- Expanded black-box transfer experiments to other editing models, including two flow-based models and a GAN-based deepfake model as targets (jjT1, fDxR)
- Additional ablation study on the adaptive blurring component (Y1wo)
- Additional quantitative and qualitative analyses on the per-region adaptation scheme (Zo4E)
- Additional results for more challenging inpainting scenarios, e.g., under mask variability (Zo4E)
- Improved clarity and flow reflecting editorial comments and clarifications (fDxR, Y1wo, jjT1, Zo4E)

We believe these updates will further strengthen our manuscript, and we are now more confident that our work can make a meaningful contribution to the NeurIPS community.

Thank you once again for your consideration.

Best regards, \
Authors

---

### Decision · Program_Chairs · 2025-09-17

**Decision:**

Accept (poster)

**Comment:**

This paper presents a simple method for protecting an image against AI-based image editing. The idea extends existing perturbation-based protection methods to local regions by utilizing segmentation masks. Experiments demonstrate more effective protection over baselines. Overall, the paper is easy to understand and well motivated. The reviewers requested several additional experiments, which the authors have addressed; hence, the reviewers are supportive of this paper. The AC agrees with this evaluation.